# RESOLUTION ATTACK: EXPLOITING IMAGE COMPRESSION TO DECEIVE DEEP NEURAL NETWORKS

**Wangjia Yu**[1,2], **Xiaomeng Fu**[1,2], **Qiao Li**[1,2], **Jizhong Han**[1], **Xiaodan Zhang**[1*]
[1]Institute of Information Engineering, Chinese Academy of Sciences
[2]School of Cyber Security, University of Chinese Academy of Sciences
{yuwangjia,fuxiaomeng,liqiao,hanjizhong,zhangxiaodan}@iie.ac.cn

## ABSTRACT

Model robustness is essential for ensuring the stability and reliability of machine learning systems. Despite extensive research on various aspects of model robustness, such as adversarial robustness and label noise robustness, the exploration of robustness towards different resolutions, remains less explored. To address this gap, we introduce a novel form of attack: the resolution attack. This attack aims to deceive both classifiers and human observers by generating images that exhibit different semantics across different resolutions. To implement the resolution attack, we propose an automated framework capable of generating dual-semantic images in a zero-shot manner. Specifically, we leverage large-scale diffusion models for their comprehensive ability to construct images and propose a staged denoising strategy to achieve a smoother transition across resolutions. Through the proposed framework, we conduct resolution attacks against various off-the-shelf classifiers. The experimental results exhibit high attack success rate, which not only validates the effectiveness of our proposed framework but also reveals the vulnerability of current classifiers towards different resolutions. Additionally, our framework, which incorporates features from two distinct objects, serves as a competitive tool for applications such as face swapping and facial camouflage. The code is available at https://github.com/ywj1/resolution-attack.

## 1 INTRODUCTION

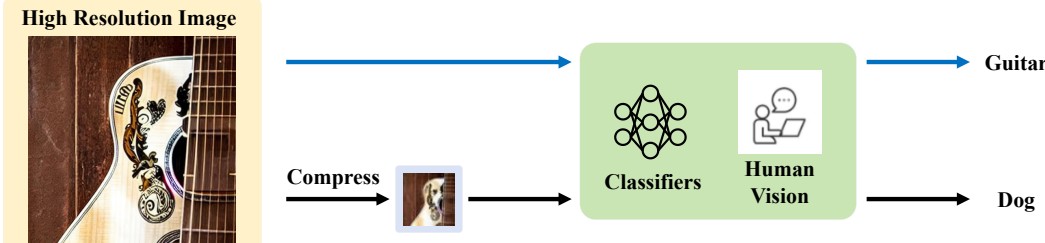

Figure 1: An Example of the proposed Resolution Attack.

Model robustness has attracted significant attention in recent research, as it focuses on the fundamental ability of a machine learning model to maintain its performance and stability in the presence of a wide array of challenges, including uncertainties stemming from data variability, disturbances in environments and even deliberate attacks aimed at compromising the model's integrity. Robustness represents a cornerstone to ensure the stability and reliability of models in practical applications, particularly in critical domains such as autonomous driving and medical diagnosis.

There exists extensive research on model robustness across diverse dimensions, such as adversarial robustness and label noise robustness. However, research on robustness across different resolutions

---

*Corresponding Author

remains underexplored. In real-world scenarios, classifiers are often exposed to images ranging from very high fidelity to severely compressed. Classifiers should exhibit robustness across these varying resolutions, particularly for low-resolution images, as many practical applications inevitably involve such inputs. For example, images on social media platforms or IoT devices are often compressed to reduce storage and transmission costs. Similarly, in autonomous driving or surveillance systems, objects that are distant or small are frequently captured in low resolution due to technological limitations. These low-resolution images may degrade semantic content and introduce inaccuracies in predictions, particularly under maliciously crafted inputs. As illustrated in Figure 1, an intricate high-resolution image (e.g., *guitar*) is initially compressed to align with the classifier's training resolution, yet the quality degradation during this process leads to misclassification, in this case, identifying the image as *dog*.

The crafted image effectively exploits a vulnerability in classifiers, successfully executing an attack by presenting dual semantic representations. Specifically, it manifests as a guitar in high resolution while resembling a dog in low resolution, thereby deceiving both classifiers and human observers. This novel attack paradigm, which we term the Resolution Attack (RA), also serves as a metric to assess the robustness of current classifiers towards inputs across various resolutions. The key to execute the resolution attack lies in the crafted image with dual semantic representations, which entails several requirements: 1) **High Resolution Fidelity**. It ensures that the image appears natural and free of perceptual noise or perturbations, faithfully conveying the semantics of the high-resolution content (e.g., *the guitar* in Figure 1). 2) **Low Resolution Misclassification**. The compressed low resolution image should successfully mislead classifier algorithms. Additionally, it must convincingly represent the semantic content at low resolutions (e.g., *the dog* in Figure 1) to deceive human observers. 3) **Consistency across Resolutions**. The image should exhibit a seamless transition between low and high resolutions, maintaining a coherent and continuous depiction across resolutions.

To meet the above requirements, we developed a resolution attack framework capable of automatically generating images featuring dual semantic representations. Leveraging the generative priors of large-scale diffusion models, we achieve the generation of high-resolution images with naturalistic and high fidelity. To meet the dual representation requirements at different resolutions, we employ a divide-and-conquer strategy. Specifically, we partition the denoising process in the frequency domain. The low-frequency is denoised to reveal the underlying semantics at a lower resolution, while the high-frequency contributed to the intricate details at higher resolutions. Subsequently, these low and high frequency are seamlessly integrated to produce the desired dual representation. Additionally, we introduce a staged strategy that focuses different semantics in different denoising stages, facilitating a smoother transition across resolutions.

Moreover, our proposed framework is capable of generating dual representation images that embed specific entities, such as celebrities or objects, within the high-resolution content. For example, our method can effectively "hiding" the given white dog within various contents, as depicted in Figure 2(b). We term this variant the Resolution Attack with Source Image. Building upon the aforementioned resolution attack, the resolution attack with source image incorporates an additional source image as input to guide the geometry of the generated image. Consequently, there is an additional requirement: the generated image should encapsulate the source image at low resolution. To address this, we incorporate inversion techniques and propose a structural guidance module, which enforces a rigorous structural alignment between the source and generated images.

We employ the proposed framework to create images with dual representations for the purpose of resolution attack. The experimental results reveal that current classifiers are vulnerable to the resolution attack. More intriguingly, due to the capability of integrating features from two distinct objects, our framework can be utilized for face swapping and facial camouflage, posing further challenges for face recognition models. Our contributions can be summarized as follows:

- We define a novel form of attack: the resolution attack. By generating images with different semantics across different resolutions, Resolution attacks effectively deceives both classifier and the human observers.

- To execute the resolution attack, we propose an automated framework capable of generating dual semantic images in a zero-shot manner. Leveraging the generative priors of large-scale diffusion models and a staged strategy, our framework achieves a seamless transition across different resolutions.

- We leverage the proposed framework to attack current classifiers. The high attack success rate validates the efficacy of our proposed framework. Besides, thanks to the capability of incorporating features from two distinct objects, our framework emerges as a viable tool for face swapping and facial camouflage.

## 2 PRELIMINARY

### 2.1 DIFFUSION MODELS

Diffusion models (Song et al., 2021) are generative models that aim to learn the inverse of a forward noise process, and consist of two processes: the diffusion (forward) process and the denoising (reverse) process. The diffusion process gradually adds noise to the data distribution, ultimately reaching a known simple distribution (often a Gaussian distribution). This process can be described as a Markov chain, iteratively introducing Gaussian noise into the original image $x_0$, with a total of $T$ steps:

$$q(x_{1:T}|x_0) = \prod_{t=1}^{T} q(x_t|x_{t-1}) \tag{1}$$

where:

$$q(x_t|x_{t-1}) = \mathcal{N}(x_t; \sqrt{1 - \beta_t}x_{t-1}, \beta_t \mathbf{I}) \tag{2}$$

and the variance schedule $\beta_1, \ldots, \beta_T$ is predefined. As $t$ approaches $T$, $\beta_t$ becomes closer to 1.

The denoising (reverse) process iteratively denoises pure white noise samples. Diffusion models begin with Gaussian noise and, through a series of denoising steps, generate images:

$$p_\theta(x_{0:T}) = p(x_T) \prod_{t=1}^{T} p_\theta(x_{t-1}|x_t) \tag{3}$$

where:

$$p_\theta(x_{t-1}|x_t) = \mathcal{N}(x_{t-1}; \mu_\theta(x_t, t), \Sigma_\theta(x_t, t)) \tag{4}$$

and $\Sigma_\theta(x_t, t)$ is a constant depending on $\beta_t$, and $\mu_\theta(x_t, t)$ is predicted by a neural network $\epsilon_\theta$ as:

$$\mu_\theta(x_t, t) = \frac{1}{\sqrt{\alpha_t}} \left( x_t - \frac{\beta_t}{\sqrt{1 - \bar{\alpha}_t}} \epsilon_\theta(x_t, t) \right) \tag{5}$$

### 2.2 DDIM INVERSION

To expedite the denoising process and ensure a unique output, deterministic DDIM sampling (Song et al., 2021) has been introduced, thereby enabling a skip-step strategy. The diffusion process, named DDIM inversion, has been suggested for the DDIM sampling. The DDIM inversion technique is a method for efficient data generation, which accelerates the generation process by introducing a deterministic non-Markovian process, while maintaining the high quality of the generated samples. Such an inversion process, shown in Equation 6, provides a deterministic transformation between an input image and its corrupted version.

$$x_{t+1} = \sqrt{\alpha_{t+1}} \left( \frac{x_t - \sqrt{1 - \alpha_t}\epsilon_\theta(x_t, t)}{\sqrt{\alpha_t}} \right) + \sqrt{1 - \alpha_{t+1}}\epsilon_\theta(x_t, t) \tag{6}$$

In our work, we employ DDIM inversion to transform an input image into the noisy latent embedding. This latent embedding is then used to generate images subjected to resolution attack.

## 3 PROBLEM FORMULATION

In this paper, we propose Resolution Attack (RA), which are designed to craft **high-resolution** images that exhibit entirely distinct semantic content in their **low-resolution** counterparts. The resolution attack, as exemplified in Figure 1, poses a significant challenge to both automated classifiers and human observers, underscoring the potential vulnerabilities inherent in image recognition systems.

We categorize the RA attack into two distinct types, each tailored to specific scenarios. The naive resolution attack, focuses solely on generating images that exhibit dual semantic representations across resolutions, without imposing additional constraints. Conversely, in scenarios requiring the generation of tailored content for specific individuals or objects, supplementary constraints on the generative process is needed. Specifically, a given source image is manipulated to exhibit the dual representation across resolutions. We specialize this form of attack as the resolution attack with source image. We will detail these two attacks in the following.

## 3.1 RESOLUTION ATTACK

Given a paired class $(\mathcal{C}_L, \mathcal{C}_H)$, the resolution attack (RA) algorithm $\mathcal{M}$ produces a high-resolution image $x$ that possesses dual semantic representations: the original $x$ belongs to $\mathcal{C}_H$, while its low-resolution counterpart $x_\downarrow$ (obtained through downsampling) is classified as $\mathcal{C}_L$. This attack not only requires $x_\downarrow$ to visually resemble $\mathcal{C}_L$ but also succeeds in deceiving the classifier $f$ into classifying $x_\downarrow$ as $\mathcal{C}_L$. The RA attack can be formulated as:

$$\mathcal{M}(\mathcal{C}_L, \mathcal{C}_H) = x$$
$$f(x) = \mathcal{C}_H \quad f(x_\downarrow) = \mathcal{C}_L \tag{7}$$

## 3.2 RESOLUTION ATTACK WITH SOURCE IMAGE

Conversely, the Resolution Attack with Source Image (RAS) aims to generate a specific output aligned with a predefined source image $I_s$. In addition to the paired input class $(\mathcal{C}_L, \mathcal{C}_H)$, an auxiliary source image $I_s$ is introduced to modulate the output content. This attach not only fulfills the dual semantic representation requirements in the RAS attack, but also strives to exhibit coherence with the source image. The RAS attack can be formulated as:

$$\mathcal{M}(\mathcal{C}_L, \mathcal{C}_H, I_s) = x$$
$$f(x) = \mathcal{C}_H \quad f(x_\downarrow) = \mathcal{C}_L$$
$$\min D(x_\downarrow, I_s) \tag{8}$$

where $D$ is the distance metric to measure the similarity between the source image $I_s$ and the downsampled image $x_\downarrow$.

## 4 METHOD

### 4.1 METHOD OVERVIEW

We categorize the attack into two distinct types: the resolution attack and the resolution attack with source image. The framework of these two attacks is illustrated in Figure 2. In the context of the RA attack, depicted in Figure 2(a), a generator is proposed to generate the dual semantic representation image. Specifically, the Dual-Stream Generative Denoising Module leverages an off-the-shelf diffusion model. Trained with large scale datasets, this diffusion model possess experienced understanding of how to construct an image. Thanks to this generative priors, the Dual-Stream Generative Denoising Module can handle simultaneously generating and integrating two distinct semantic contents. Conversely, the RAS attack introduces (Figure 2(b)) an additional requirement: the synthesized image must closely resemble the input source image $I_s$. To achieve this, we introduce the DDIM Inversion technique and a novel module: the Structural Guidance Module. The former serves to mitigate the randomness introduced by the initial noise, while the latter is designed to more accurately capture the structural attributes of the source image. A detailed description of these technologies and modules will be provided in the following.

### 4.2 RESOLUTION ATTACK

The RA attack is designed to produce images that exhibit distinct semantic contents across varying resolutions. It confronts several notable challenges. First and the most significantly, it must adeptly blend two contents, possessing vastly different appearances and shapes, within a single image while simultaneously maintaining the visual quality of each content to achieve a smooth conversion from

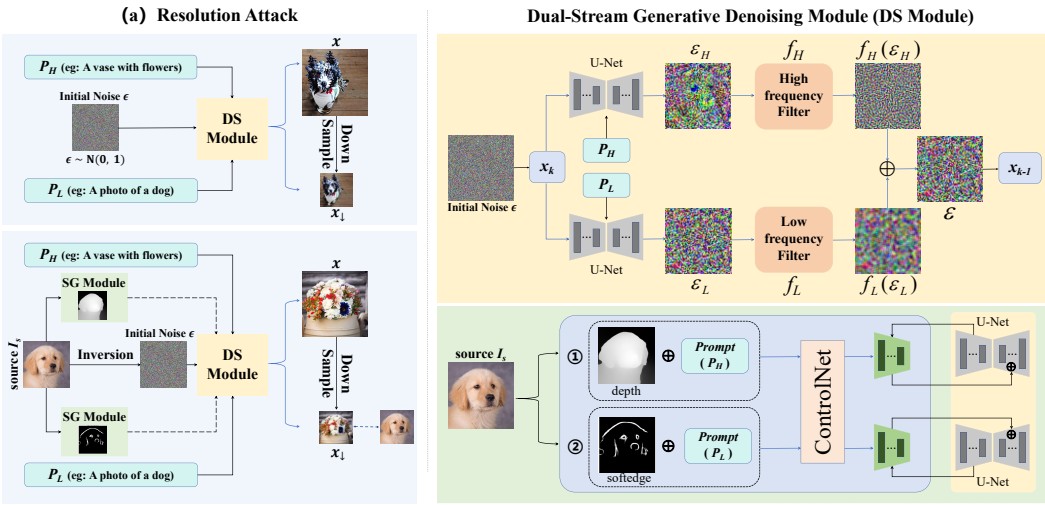

Figure 2: Overview of the RA and the RAS attack. The figure illustrates the method of attack, including the RA and the RAS attack approaches. Key components include the Dual-Stream Generative Denoising Module and the Structural Guidance Module, which process noise, source images, and prompts to generate dual semantic representation outputs.

one resolution to another. Furthermore, the capability to generate high-resolution images is also of great significance, given that images of higher resolutions inherently possess increased redundancy, allowing for the semantic content of distinct elements to be intricately embedded within this redundant pixel space. Specifically, maintaining fine-grained details related to the high-resolution class $(\mathcal{C}_H)$, while also ensuring that the broader, low-frequency characteristics of the low-resolution class $(\mathcal{C}_L)$ are preserved, requires precise control over the frequency components. To achieve this, we propose the the Dual-Stream Generative Denoising Module, which leverages the profound generative priors of a large scale diffusion model.

**The Dual-Stream Generative Denoising Module.** The Dual-Stream Generative Denoising Module (DS Module) is composed of the U-Net (Ronneberger et al., 2015) of a pretrained diffusion model, a high frequency filter and a low frequency filter. By iteratively perform the denoising step, the DS module gradually convert the initial noise $\epsilon$ into an image with dual semantic representations. In each denoising step, the prompt representing the high and low resolution class are fed into the U-Net separately to obtain corresponding noise $\epsilon_H$ and $\epsilon_L$. To ensure that the generated high-resolution image accurately represents $\mathcal{C}_H$ and the downsampled image faithfully represents the $\mathcal{C}_L$, we manipulate the output noise in the frequency domain. Concretely, the final noise is composed of two parts: the low frequency part and the high frequency part. The low frequency part comes from the $\epsilon_L$ while the rest high frequency component comes from $\epsilon_H$. In this way, the generated image will take both $\mathcal{C}_L$ and $\mathcal{C}_H$ into account. This process can be represented as:

$$\epsilon_L = \epsilon_\theta(P_L) \quad \epsilon_H = \epsilon_\theta(P_H) \tag{9}$$

$$\epsilon = f_L(\epsilon_L) + f_H(\epsilon_H) \tag{10}$$

where $P_H$ and $P_L$ is the corresponding text prompts to the class input $\mathcal{C}_H$ and $\mathcal{C}_L$. For example, the text prompts for the class dog is "*a photo of a dog*".

**Time-Dependent Denoising Strategy.** To effectively generate images with dual semantic representations while maintaining continuity between $\mathcal{C}_H$ and $\mathcal{C}_L$, we further propose the Time-Dependent Denoising Strategy. Previous researches (Choi et al., 2022; Kwon et al., 2023; Park et al., 2024; Wang et al., 2024) have revealed that diffusion models first synthesize the coarse structure (corresponding to $\mathcal{C}_L$), then the finer details are established (corresponding to $\mathcal{C}_H$). Inspired by these insights, we design the time-dependent denoising strategy that comprises three stages: early, middle, and late. In the early and late stage, we drop the dual-stream denoising process and solely utilize $\mathcal{P}_L$ or $\mathcal{P}_H$, respectively. Specifically, in the early stage, diffusion models focus on constructing the

structure which corresponds to $\mathcal{C}_L$, thus requires only $\mathcal{P}_L$. Conversely, in the late stage, we only employ $\mathcal{P}_H$ to establish finer details. The intervening middle stage leverages the dual-stream denoising process that incorporates both $\mathcal{P}_L$ and $\mathcal{P}_H$. By focusing on different semantics in different denoising stages, we facilitate a smoother transition across resolutions.

### 4.3 RESOLUTION ATTACK WITH SOURCE IMAGE

In certain applications, there is a requirement to synthesize images that are tailored to a specific individual or object. To address this limitation, we introduce an additional source image to further exert the content of the generated images. This modified approach, termed the RAS attack, aims to ensure that the synthesized images closely resemble the input source image, achieving a higher degree of control in the image generation process.

**The Inversion technology.** Previous research (Wu et al., 2025; Zhao et al., 2024) have demonstrated the pivotal role of the initial noise distribution in shaping the outcomes of the diffusion models, particularly with respect to the layout, structural composition and form of the synthesized content. Consequently, we harness the DDIM inversion (Song et al., 2021) to map the source image back to its corresponding initial noise, denoted as $\epsilon_\theta$. In the RAS attack, we substitute the random initial noise with $\epsilon_\theta$ as the initialization point for the denoising module. This strategy not only mitigates the randomness introduced by the initial noise, but also ensures that the structural attributes of the source image are preserved throughout the generation process. Furthermore, it establishes a foundation for subsequent image synthesis to produce outputs that maintain structural coherence with the input source image.

**The Structural Guidance Module.** To further preserve the structure of the source image throughout the generation process, we introduce the Structural Guidance Module (SG Module). This module integrates supplementary guidance in the forms of two distinct structural signals (Zhang et al., 2023): depth maps and softedge images. By leveraging these signals, the SG module achieves a fine-grained level of manipulation over the structural aspects of the synthesized images. The depth image provides coarse structural guidance. As an overall contour map of the source image, the depth map captures the fundamental geometric characteristics. By integrating the depth map into the image generation process, we ensure that the coarse structure of the source image is preserved in the synthesized output. In contrast, the softedge image offers more precise shape guidance, focusing on the exact boundaries and contours of the source image, ensuring that the generated images are precisely aligned with the structural and shape characteristics of the source image.

We applied the depth map and softedge image in different stages of noise prediction, experimenting with various combinations. Based on experimental results, when the semantic gap between the high-resolution prompt and the low-resolution prompt is relatively small, the best results are achieved when integrating the depth map into the high-resolution noise prediction process and the softedge image into the low-resolution noise prediction process. In contrast, when the semantic gap between the high-resolution and low-resolution prompts is larger, control maps derived from the source image may overly constrain the high-resolution image generation process, hindering its natural semantic expression. It is more effective to stop controlling the high-resolution noise prediction process and only integrate the softedge image into the low-resolution noise prediction process. The combined use of depth and softedge maps offers a balanced approach to controlling the image generation process. It ensures that the generated image not only reflects the corresponding high and low resolution classes but also preserves the structural integrity of the source image, thereby achieving a harmonious synthesis of resolution specific details and structural coherence.

## 5 EXPERIMENT

### 5.1 EXPERIMENTAL SETUP

**Target Classifiers and Datasets.** We select a range of widely utilized classifiers to evaluate the efficacy of our proposed resolution attack. These classifiers include ResNet-50 ($224\times224$) (He et al., 2016), VGG19 ($224\times224$) (Simonyan & Zisserman, 2015), InceptionV3 ($299\times299$) (Szegedy et al., 2016), EfficientNet ($384\times384$) (Tan & Le, 2021), and DenseNet ($224\times224$) (Huang et al., 2017). All these classifiers are trained on the ImageNet (Russakovsky et al., 2015) dataset, which consists of over a million images across 1000 categories.

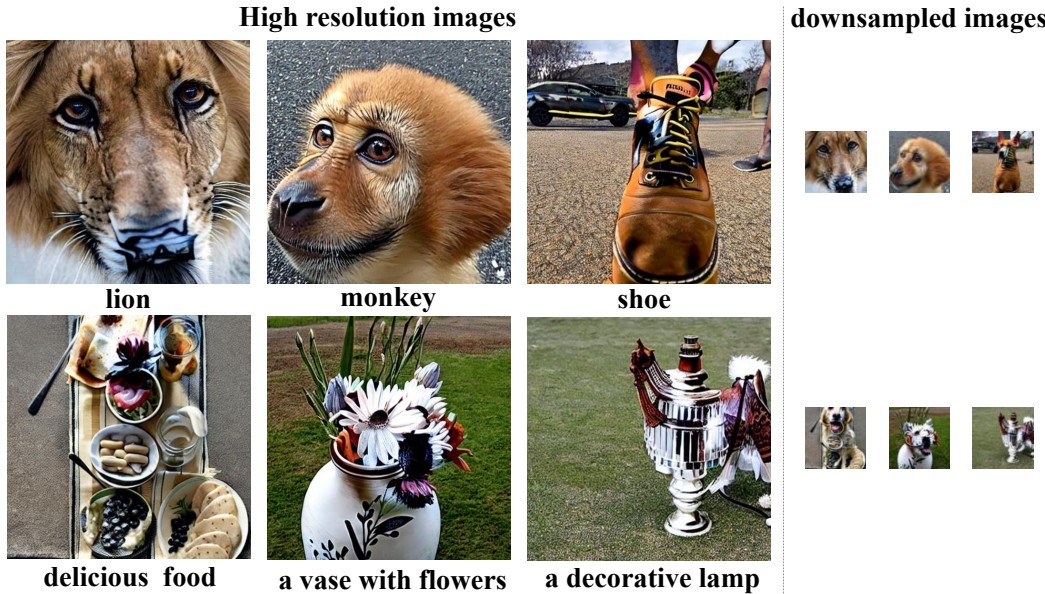

Figure 3: **Qualitative Results of RA.** Top: Labeled attacks using different high-resolution categories, including "*lion*", "*monkey*" and "*shoe*". Bottom: Unlabeled attacks using various high-resolution prompts, such as "*delicious food*", "*a vase with flowers*" and "*a decorative lamp*". On the right are the downsampled images correspond to the low-resolution outputs. We suggest further zooming in for better details or refer to the higher resolution version provided in the Appendix.

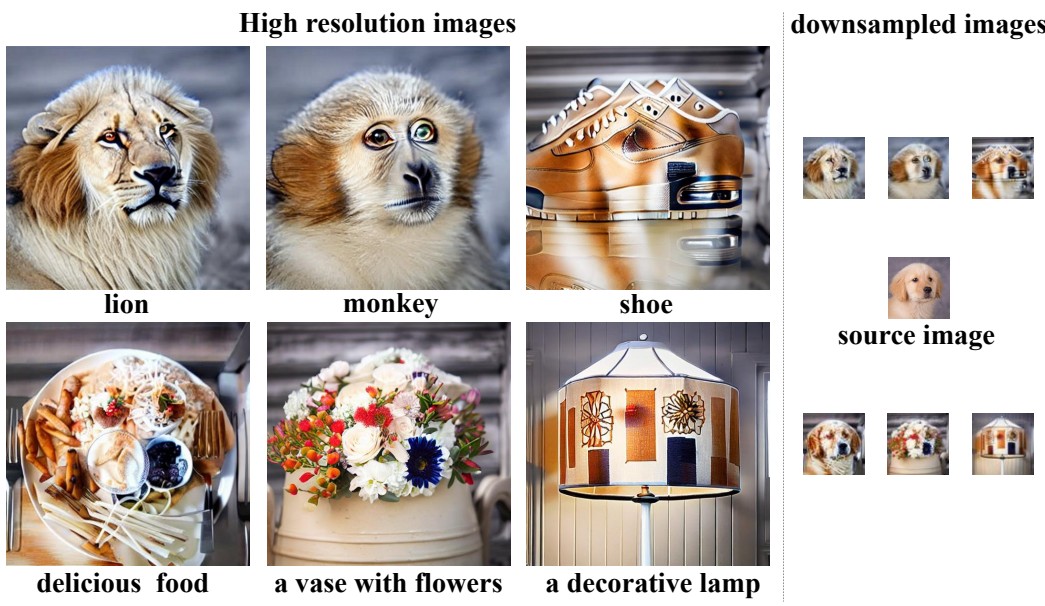

Figure 4: **Qualitative Results of RAS.** Top: Labeled attacks using different high-resolution categories, including "*lion*", "*monkey*" and "*shoe*". Bottom: Unlabeled attacks using various high-resolution prompts, such as "*delicious food*", "*a vase with flowers*" and "*a decorative lamp*". On the right are the downsampled images correspond to the low-resolution outputs, alongside the source image. We suggest further zooming in for better details or refer to the higher resolution version provided in the Appendix.

**Labeled and Unlabeled Attack.** To align with the classification capabilities of the aforementioned target classifiers, the input class pairs $(\mathcal{C}_L, \mathcal{C}_H)$ are confined to the categories present within the

Table 1: The quantitative results of the Resolution Attack.

| Classifiers | Labeled Attack | | | | | Unlabeled Attack | | |
|---|---|---|---|---|---|---|---|---|
| | $\text{Acc}_H \uparrow$ | $\text{Acc}_L \uparrow$ | $\text{ASR}_C \uparrow$ | $\text{CLIP}_H \uparrow$ | $\text{CLIP}_L \uparrow$ | $\text{Acc}_L \uparrow$ | $\text{CLIP}_H \uparrow$ | $\text{CLIP}_L \uparrow$ |
| Resnet-50 | 68.5% | 71.8% | 71.4% | | | 63.2% | | |
| VGG19 | 65.3% | 64.8% | 61.6% | | | 59.5% | | |
| InceptionV3 | 76.7% | 43.3% | 43.6% | 0.298 | 0.248 | 26.0% | 0.256 | 0.247 |
| EfficientNet | 89.6% | 67.3% | 68.2% | | | 42.0% | | |
| DenseNet | 69.8% | 63.4% | 60.5% | | | 69.3% | | |

Table 2: The quantitative results of the Resolution Attack with Source image.

| Classifiers | Labeled Attack | | | | | | Unlabeled Attack | | | |
|---|---|---|---|---|---|---|---|---|---|---|
| | $\text{Acc}_H \uparrow$ | $\text{Acc}_L \uparrow$ | $\text{ASR}_C \uparrow$ | $\text{CLIP}_H \uparrow$ | $\text{CLIP}_L \uparrow$ | SSIM$\uparrow$ | $\text{Acc}_L \uparrow$ | $\text{CLIP}_H \uparrow$ | $\text{CLIP}_L \uparrow$ | SSIM$\uparrow$ |
| Resnet-50 | 59.5% | 78.6% | 77.1% | | | | 38.8% | | | |
| VGG19 | 58.1% | 77.1% | 74.9% | | | | 40.0% | | | |
| InceptionV3 | 61.3% | 46.0% | 42.6% | 0.295 | 0.247 | 0.727 | 18.9% | 0.266 | 0.217 | 0.660 |
| EfficientNet | 79.9% | 70.7% | 70.3% | | | | 30.5% | | | |
| DenseNet | 57.4% | 77.5% | 75.3% | | | | 44.0% | | | |

ImageNet dataset. We refer to this type of attack as the **labeled attack**, as the class pairs fall within the scope of the classifiers' training label set. Nonetheless, we claim that our attack can produce dual semantic representations beyond the categories of the training labels. To demonstrate this, we select more abstract concepts as the high-resolution class $\mathcal{C}_H$, e.g. "*a photo of a calm lake*" or "*a photo of a delicious plate of food*". We maintain $\mathcal{C}_L$ within the label sets to test the attack success rate. We refer to this type of attack as the **unlabeled attack**.

**Evaluation Metrics.** We evaluate the effectiveness of our attack in following aspects as described in Equation 7 and 8: **High/Low-Resolution Classification Accuracy** ($\text{Acc}_H$/$\text{Acc}_L$): The accuracy of the target classifiers for the high-resolution images $x$ and low-resolution images $x_\downarrow$. Specifically, we generate high-resolution images with a resolution of $512\times512$ pixels and downsample them by a factor of 3 to obtain low-resolution images with a resolution of $64\times64$ pixels. This experimental setup is designed to simulate real-world scenarios where classifiers process low-resolution inputs. When computing accuracy, both high-resolution and low-resolution images are directly fed into the target classifiers. The classifiers preprocess these images using PyTorch's default pipeline[1] to ensure alignment with the required input dimensions. **Corrective Attack Success Rate** ($\text{ASR}_C$): The proportion of cases where both the high-resolution image $x$ and its low-resolution counterpart $x_\downarrow$ are classified as described by their corresponding prompt classes. This metric highlights the vulnerability of classifiers across resolutions. **CLIP Score** ($\text{CLIP}_H$/$\text{CLIP}_L$): The cosine similarity between the text and image embeddings by CLIP's encoder (Radford et al., 2021) for high/low-resolution images. **SSIM**: As the RAS requires that the low-resolution image $x_\downarrow$ resemble the source image $I_s$, we evaluate this similarity using the SSIM (Wang et al., 2004) between $I_s$ and $x_\downarrow$.

**Implementation Details.** We utilize the Stable Diffusion (Rombach et al., 2022) v1.5 as the denoising U-Net within the Dual-Stream Generative Denoising Module. We iteratively generate the dual-representation images over 300 denoising steps, employing a time-dependent strategy where the first 20 steps utilize the $\mathcal{C}_L$, the last 20 steps utilize the $\mathcal{C}_H$ and the 260 steps in between utilize both $\mathcal{C}_L$ and $\mathcal{C}_H$. Gaussian Filters are employed as the low-frequency filter, while the high-frequency filter is constructed by substracting the low-frequency component extracted by the Gaussian Filters. For RAS, we utilize the ddim inversion of 200 steps to embed the source image into the latent noise space. Notably, a source image is required for RAS. For this purpose, we collect a dataset of 100 frontal images of dogs as the source images. We give more information about the implementation details in Appendix A.

---

[1]Specifically, PyTorch preprocesses the input image of varying resolutions to match the required resolution of the target classifier. For instance, ResNet first resizes then central crops the image into 224x224. More information can be found in the PyTorch official document.

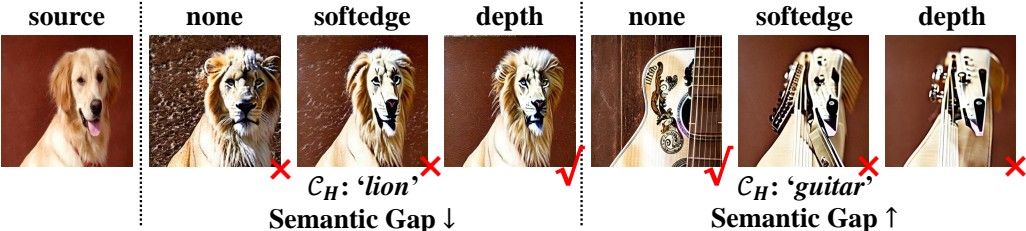

Figure 5: Different Structure Control with different semantic gaps. When the gap is small (left), we leverage the depth map as the structural guidance. When the gap is large, none guidance is used to ensure the expression of the high-resolution class $\mathcal{C}_H$.

## 5.2 EXPERIMENTAL RESULT

**Qualitative Results.** Figure 3 and Figure 4 present qualitative examples of RA and RAS respectively. It can be observed that the generated images exhibit clear dual semantic representations across resolutions. Furthermore, in the context of RAS (Figure 4), the downsampled images adhere to the $\mathcal{C}_L$ class label while concurrently preserving structural fidelity with the source image. This observation underscores the efficacy of integrating the inversion process and the proposed SG module. Due to the space constraints, the presented high-resolution images have been compressed to accommodate a greater number of generated images. For a more detailed view, we encourage readers to zoom in on these images or refer to the images provided in the Appendix F for enhanced clarity.

**Quantitative Evaluation.** Table 1 and 2 exhibit the quantitative results of RA and RAS towards various classifiers. Our findings indicate that all classifiers are susceptible to the proposed resolution attack. Furthermore, a comparison reveals that labeled attacks outperform unlabeled attacks in presenting low-resolution semantics (higher $\text{Acc}_L$), This is attributed to the fact that unlabeled attack leverage more abstract concepts as the $\mathcal{C}_H$. The abstract representations in the high-resolution images are mirrored in the low-resolution images, which in turn leads to a decrease in $\text{Acc}_L$ metric. Besides, we observe that both attacks demonstrate impressive CLIP score, with the optimal score reaching 0.298, which further validate that our method can generate dual semantic representations with excellent consistency across resolutions. Additional, thanks to the proposed SG module, the generated images can accurately represent the structure of the source image, achieving SSIM values of 0.727 and 0.660, respectively.

## 5.3 ABLATION STUDY

**The Structure Guidance Module.** We employ both softedge and depth maps to obtain finer control over the structure of the generated images. For better comprehension of the characteristics of these two maps, we provide specific examples in Figure 5. We fix the low-resolution structural guidance as the softedge and examine how different high-resolution structural guidance impacts the generated images. We examine two distinct scenarios: one with small semantic gaps (e.g., *dog* and *lion*) and another with large gaps (e.g., *dog* and *guitar*). When the gap is small, the absence of control (none) results in the model neglecting the source image's structure. The softedge is another extreme case where too fine-grained structure control is provided that the generated image closely resemble the source image but overlook the high-resolution class $\mathcal{C}_H$. The depth map offers appropriate structural guidance, enabling successful generation of dual-semantic images. However, when the semantic gap is large, applying structural control like softedge and depth maps impede the expression of $\mathcal{C}_H$, leading to unsatisfactory results. Therefore, the semantic gap is of great significance when leveraging structural guidance.

## 5.4 RESOLUTION ATTACKS AS THE FACE SWAPPER

Our proposed Resolution Attacks possess the capability to generate images that exhibit dual representations, incorporating features from two distinct objects within a single image. This capacity can further be harnessed for implicit face swapping by blending facial features from different identities.

The results are depicted in Figure 6, while an enhanced clarity version is provided in the Appendix F. Additionally, our Resolution Attacks can be employed for facial camouflage (e.g., "hiding" a face in flowers). The facial camouflage results are provided in the Appendix C.

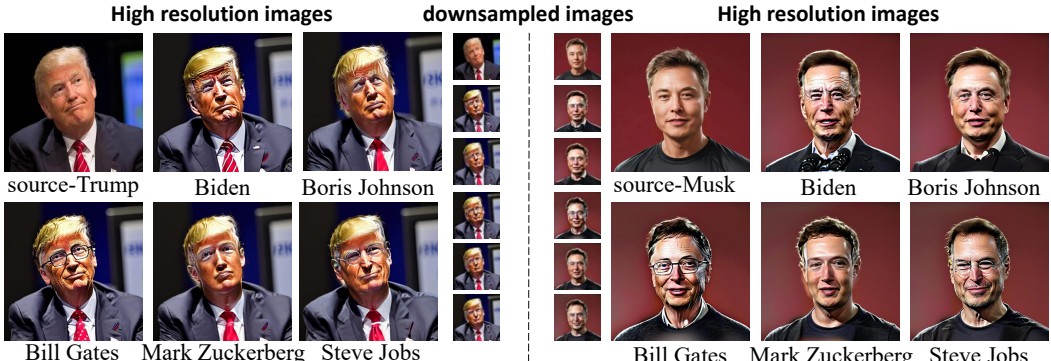

Figure 6: **Resolution attacks as the face swapper.** High-resolution images are generated using *Donald Trump* (left) and *Elon Musk* (right) as the source image, combined with various prominent figures such as *Biden*, *Boris Johnson*, *Bill Gates*, *Mark Zuckerberg*, and *Steve Jobs*. The middle column shows the corresponding downsampled low-resolution images.

## 6    CONCLUSION

In this paper, we investigate the vulnerability of contemporary classifiers to variations in image resolutions, particularly in high-resolution images and their compressed counterparts. We define a novel form of attack, termed the "resolution attack", which deceives both classifiers and human observers by producing images that exhibit distinct semantics across different resolutions. Furthermore, an automated framework is contributed to execute this resolution attack. Leveraging the generative priors of large-scale diffusion models, the framework can generate images with dual semantics in a zero-shot manner. Considering that the ability of generative models is becoming stronger and stronger, we hope that our work can serve as an alarm to possible future attacks. We anticipate that the resolution attack will stimulate further research into resolution robustness, and the proposed framework will contribute to advancements in this area.

### ETHICS STATEMENT

Our work aims to explore vulnerabilities in machine learning classifiers through a novel paradigm of resolution attacks. This research is not designed for malicious purposes; rather, it serves as a tool to evaluate and address potential weaknesses in machine learning systems, with the ultimate goal of inspiring further research and developing more secure, reliable models. However, similar to other AI image generation technologies, our method could potentially be misused for applications such as face swapping or digital camouflage, and we unequivocally condemn any actions intended to produce harmful content. To mitigate potential risks, we propose methods that can reduce such vulnerabilities— for instance, the images generated by our approach can be used to fine-tune existing classifiers, thereby improving their robustness against resolution-based attacks. Moreover, since our generated images are produced via a stable diffusion process rather than being natural images, they can be detected using deepfake detection systems. Overall, our research aims to advance the understanding of model robustness while promoting responsible and ethical use of these techniques in the broader research community.

### ACKNOWLEDGMENTS

This project is supported by the National Key Research and Development Program of China (No.2022YFB2702500).

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

## A    IMPLEMENTATION DETAILS

The image generation process in our experiments is handled by the Stable Diffusion (Rombach et al., 2022) v1.5 model, which is trained on the LAION-5B dataset (Schuhmann et al., 2022). This model excels at generating high-quality, high-resolution images from textual descriptions, making it well-suited for our attack. In the experiment, we generate images at a fixed resolution of 512x512, using the DDIM (Song et al., 2021) sampling technique. The sampling process consists of 300 steps, ensuring sufficient time for image refinement and convergence. In the RAS Attack, DDIM inversion (Song et al., 2021) is performed over 200 steps to accurately reconstruct and guide the generation process based on the source image. During the generation process, we utilize classifier-free guidance (CFG) (Ho & Salimans, 2021) to control the fidelity and diversity of the images. Specifically, the CFG coefficient for the high-resolution prompt ($P_H$) is set to 9, while for the low-resolution prompt ($P_L$), the CFG coefficient is set to 7. Additionally, the generated images are downsampled by a factor of 3 (downsampling to 64 resolution) for the low-resolution attack evaluation. This compression process ensures that the low-resolution images differ significantly in visual quality compared to the high-resolution ones, facilitating the dual representation attack. To further

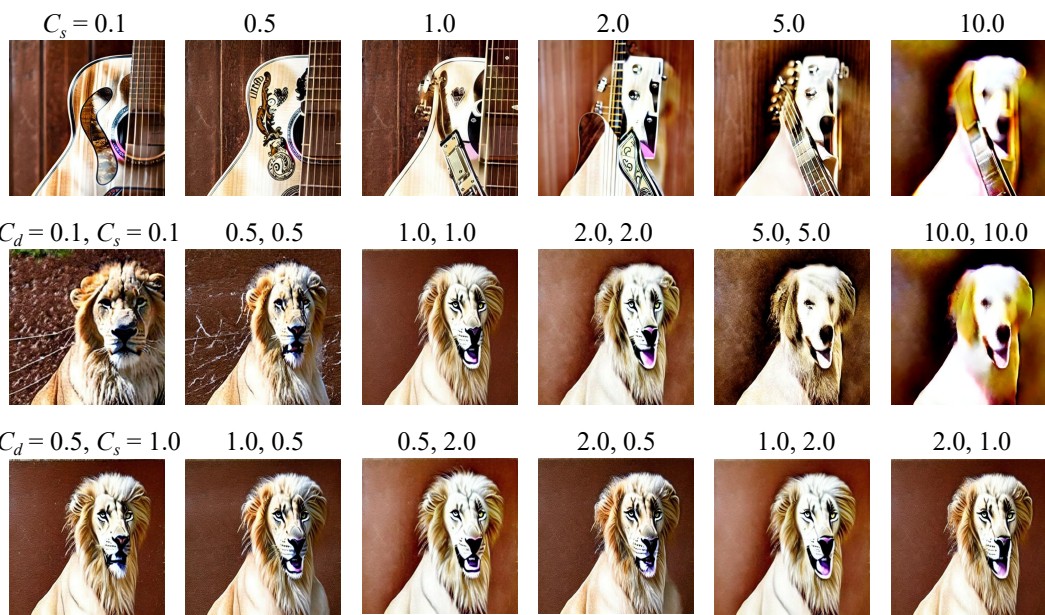

Figure 7: **Ablation on Control Parameter Settings.** Top: Effects of different control parameters in labeled object attacks. Middle and Bottom: Effects of different control parameters in labeled animal attacks. $C_d$ represents the control parameter for the depth image, and $C_s$ represents the control parameter for the softedge image.

manipulate resolution-specific features, we apply Gaussian filters during the image generation process. High frequency Gaussian filter is used to extract high-frequency details from noise (based on the prediction for the $P_H$) and low frequency Gaussian filter is applied to extract low-frequency information from noise (based on the prediction for the $P_L$), which ensures the generation of images with clearly distinct high-resolution and low-resolution features.

To enhance the generation of images with dual semantic representations, we adopt a Time-Dependent Denoising Strategy. The denoising process is divided into three phases: the early phase (first 20 steps), the middle phase (middle 260 steps), and the late phase (last 20 steps). This phased approach promotes the smooth generation of dual semantic images and ensures a gradual transition between high-resolution and low-resolution features. For both the labeled and unlabeled attacks, we generate images using the unified $P_L$ prompt along with 10 distinct $P_H$ prompts. Each prompt combination generates 100 images, with a total of 1,000 images for the labeled attack and another 1,000 images for the unlabeled attack. The RAS attack builds upon the RA attack by introducing an additional source image to guide the image generation process via DDIM inversion technology and the SG module. Similar to the RA attack, the RAS experiments are divided into labeled and unlabeled categories, and the prompts for $P_H$ and $P_L$ remain consistent. For labeled attacks involving animals, during the noise prediction stage, the depth map of the source image is used to control the noise prediction when the $P_H$ prompt is used to predict the noise, and the soft edge of the source image is used to control the noise prediction when the $P_L$ prompt is used to predict the noise. This ensures that the high-resolution features adhere to the overall structure of the source image, while the low-resolution image retains the detailed structural alignment with the source. For labeled attacks involving objects and unlabeled attacks, we apply the the SG module only during the $P_L$ prompt's noise prediction. The $P_H$ prompts differ too much semantically from the $P_L$ prompts, and applying the SG module during the noise prediction of the $P_H$ prompts may lead to poor image generation. The additional structural control could over-constrain the image generation process, hindering the natural representation of the $P_H$ prompts.

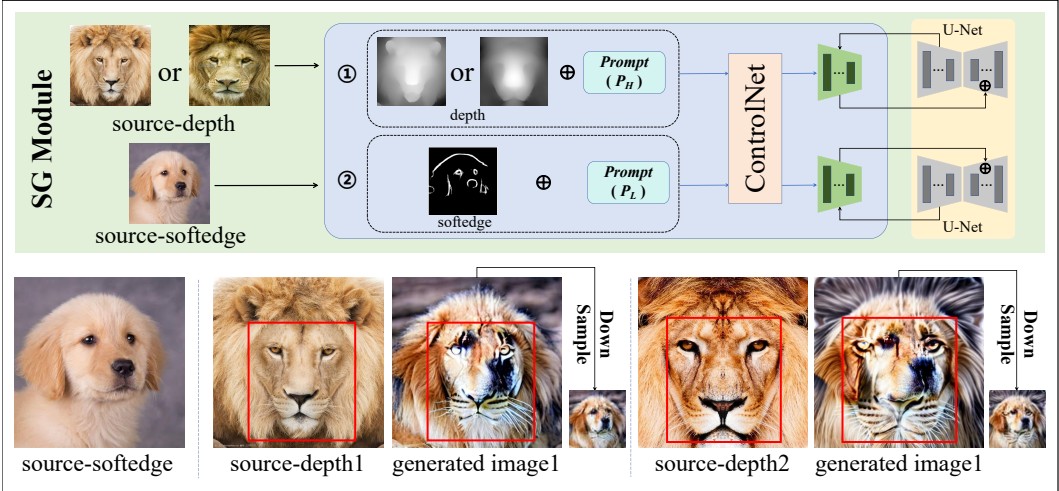

Figure 8: **Ablation on Multiple Source Images.** We provide an additional source image corresponding to the high-resolution class label $\mathcal{C}_\mathcal{H}$ (the lion image).Notably, the high-resolution images exhibit lion-structured artifacts (marked by red boxes).

Table 3: The Corrective Attack Success Rate ($\text{ASR}_C$) for labeled RAS at the 768 resolution v.s. 512 resolution. $\Delta$ denotes the improvements achieved by increasing resolution from 512 to 768.

| Resolution | ResNet-50 | VGG19 | InceptionV3 | EfficientNet | DenseNet |
|---|---|---|---|---|---|
| 512×512 | 71.4% | 61.6% | 43.6% | 68.2% | 60.5% |
| 768×768 | 77.5% | 74.0% | 68.9% | 81.4% | 78.8% |
| $\Delta \uparrow$ | **6.1%** | **12.4%** | **25.3%** | **13.2%** | **18.3%** |

# B    ADDITIONAL ABLATION STUDY

**Control parameters in the SG Module.** In the SG Module, the control parameter $C$ is used to adjust the degree to which the depth ($C_d$) and softedge ($C_s$) images influence the generated images. As shown in Figure 7, when the control parameter is small, the similarity between the generated image and the source image decreases. As the control parameter increases, the generated images become more aligned with the source image in terms of structural details. However, when the control parameter becomes too large, it negatively affects the overall quality of the generated images, introducing artifacts and potentially compromising the resolution attack effectiveness. Balancing the control parameter is thus crucial for generating high-quality images that maintain both high-resolution and low-resolution characteristics, without introducing excessive distortion.

**The Resolution.** We generate images with dual representations in 512×512 resolution. In this experiment, we explore an increasing resolution (i.e., 768×768). The Corrective Attack Success Rate ($\text{ASR}_C$) for labeled RAS at the 768 resolution is presented in Table 3. Our findings indicate that generating images with higher resolutions results in improved attack performance. This is intuitive, as higher resolutions provide additional redundancy, enabling the manifestation of more intricate dual semantic representations.

**Sampling Step.** In this experiment, we use the DDIM sampling method with different denoising sampling steps: 50, 150, 300, 500, 700, and 999. As shown in Figure 10, several intriguing observations emerges. First, the results generated with 150 steps are quite similar to those generated with 300 steps, both in terms of visual quality and structural consistency. Additionally, the images generated with 50, 500, and 999 sampling steps display similar characteristics, with comparable detail and overall image quality. Interestingly, the images generated using 700 sampling steps exhibit more detailed visual features, suggesting a potential sweet spot for retaining finer details in the generated output. These observations raise interesting questions about how different sampling steps influence

**High resolution images**    **downsampled images**

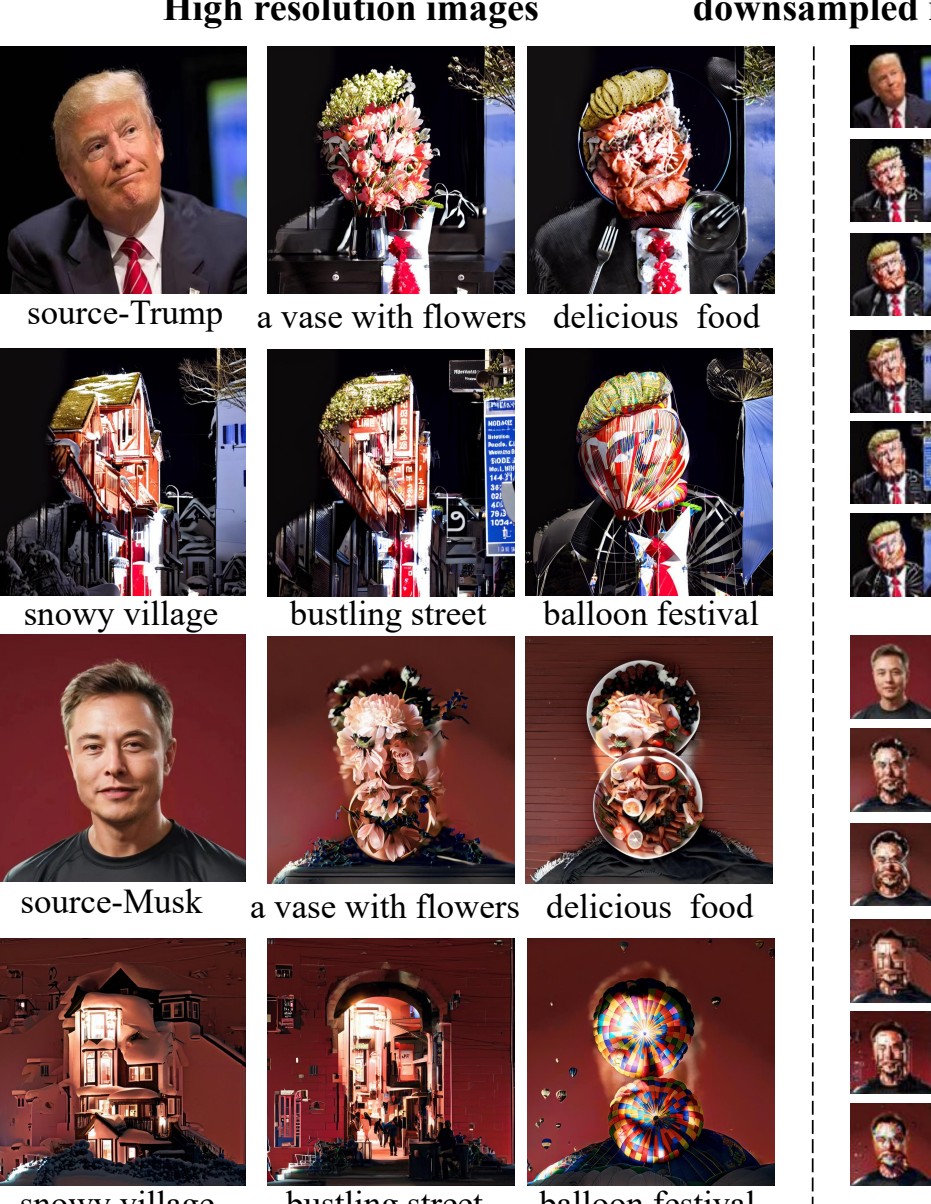

Figure 9: **Scenes-to-Face Resolution Attacks.** On the left side, high-resolution images are generated using *Donald Trump* as the source image, integrated with various scenes such as *a vase with flowers*, *delicious food*, *snowy village*, *bustling street*, and *balloon festival*. The right side presents high-resolution images created with *Elon Musk* as the source image, similarly combined with the same scenes. The middle column displays the corresponding downsampled low-resolution images.

the generative process in diffusion models. Further investigation into the relationship between sampling steps and image quality could lead to deeper insights into the workings of diffusion models and help optimize generation quality while maintaining attack effectiveness.

**Multiple Source Images.** In previous RAS experiments, both the depth and softedge maps are derived from a single source image. In this experiment, we introduce an additional source image corresponding to the high-resolution class label $\mathcal{C}_H$ (the lion in Figure 8). As shown in Figure 8, the generated images integrate features from both the original source image and the additional lion

image. Notably, the high-resolution images exhibit lion-structured artifacts (especially in the mouth area). This may be attributed to potential conflicts between the source images (the lion and dog), leading to unsatisfactory outcomes.

## C  RESOLUTION ATTACKS FOR FACIAL CAMOUFLAGE

Our proposed Resolution Attacks possess the capability to generate images that exhibit dual representations, incorporating features from two distinct objects within a single image. In this experiment, we extend the resolution attack framework by combining facial features with objects and scenes to generate facial camouflages. As illustrated in Figure 9, we generate high-resolution images that blend faces (e.g., *Donald Trump*, *Elon Musk*) with distinct scenes (e.g., *a vase with flowers*, *snowy village*). These high-resolution images exhibit detailed visual details, capturing both the facial and contextual characteristics.

## D  RELATED WORK

### D.1  MODEL ROBUSTNESS

Model robustness refers to the ability of a model to maintain its performance and accuracy in the presence of various perturbations, such as input noise, adversarial attacks, or changes in data distribution. Model robustness is crucial as it ensures the reliability and stability of machine learning systems in real-world applications. Extensive research has been conducted for model robustness. For instance, the adversarial robustness (Carlini & Wagner, 2017; Goodfellow et al., 2015) aims to boost the resilience of models towards small, intentional adversarial perturbations in the input image. Research on label noise (Natarajan et al., 2013; Li et al., 2020) focuses on developing models that are resilient to training data with incorrect or ambiguous labels. However, existing research primarily focuses on robustness to adversarial or noisy perturbations, neglecting an equally significant aspect: robustness across image resolutions. With advancements in generative models, the production of ultra-high-resolution images is now feasible without significant barriers. Therefore, it is of great significance to explore the robustness towards ultra-high resolutions. To address this, we define the resolution attack. which serves as a metric to quantify vulnerability across resolutions.

### D.2  DIFFUSION MODELS

Diffusion models have seen rapid development in recent years, becoming a focal point of current research and demonstrating significant progress across various applications (Dhariwal & Nichol, 2021; Liu et al., 2023; Ramesh et al., 2022; Ruiz et al., 2023; Saharia et al., 2022). The primary training objective of diffusion models is to iteratively add Gaussian noise to data and learn how to reverse this process by denoising the noisy data to recover the original signal. In text-to-image diffusion models (Rombach et al., 2022), image generation is treated as an iterative denoising process guided by text prompts, typically within an image encoder-decoder framework. These models utilize text embeddings generated by a language encoder (Raffel et al., 2020) as conditions, performing denoising in the latent space (Gu et al., 2022; Podell et al., 2024), and subsequently reconstructing the latent samples back into pixel space via a decoder. Furthermore, text-to-image diffusion models are capable of capturing high-level semantic concepts (Li et al., 2023a; Xu et al., 2023). Our work builds on this characteristic of diffusion models to explore the generation of dual semantic representations.

### D.3  ADVERSARIAL EXAMPLES

Adversarial examples (Goodfellow et al., 2015; Madry et al., 2017; Ilyas et al., 2019; Kurakin et al., 2017) are carefully crafted inputs to machine learning models that have been intentionally designed to cause the model to make a mistake. These inputs are typically created by introducing small, often imperceptible perturbations to the original data, which are specifically calculated to exploit the model's decision boundaries. The resulting perturbed inputs, while indistinguishable from the original to the human eye, lead to misclassifications or erroneous outputs when processed by the model.

**Diffusion Models for Adversarial Examples.** With the swift development of diffusion models, several recent studies have leveraged the generative priors of the diffusion models to craft adversarial examples. Xue et al. (2023) specifically align adversarial examples with natural images using diffusion models. Wang et al. (2023) utilizes diffusion models to learn perturbations of the latent space to create adversarial imagery. Chen et al. (2024) leverages diffusion models to manipulate the texture of the target images to compose adversarial examples.

However, adversarial examples exhibit significant limitations. First, adversarial examples are model-specific. The generation of adversarial examples heavily relies on the specific architecture and training data of the target model, meaning that adversarial examples crafted for one model may not effectively transfer to other models, even if they share similar tasks or architectures. This limits the generalizability of adversarial examples. Second, adversarial examples are typically based on pixel-level perturbations within a narrow p-ball of the target image. Considering the proliferation of various generated models on the internet and the increasing number of generated images faced by traditional classifiers, this narrow focus is problematic. In contrast, our proposed resolution attack is both model-agnostic and task-agnostic, exhibiting superior generalized performance across various architectures and tasks. By carefully crafting the semantics of the target images, the Resolution Attack can deceive not only classifiers but also human vision.

## E  MORE QUANTITATIVE RESULTS

Description and analysis to evaluate the effectiveness of resolution attacks (RA and RAS) across diverse model architectures, we conducted experiments on seven models grouped into three distinct categories:

- **Transformer-Based Models:** We employed two Vision Transformer (ViT) variants (Dosovitskiy et al., 2021), ViT-b32 and ViT-l32, which are commonly recognized for their robust image classification capabilities.

- **Feature Pyramid-Based Models:** Two object detection models, fasterrcnn_resnet50_fpn and maskrcnn_resnet50_fpn, were tested. Both models leverage ResNet50 as their backbone and incorporate feature pyramid structures (Lin et al., 2017) to handle multi-scale information effectively. Trained on the COCO dataset (containing 80 categories), these models were evaluated only on their low-resolution attack success rate, as the categories of our high-resolution images may not fully align with the COCO dataset.

- **Vision-Language Models (VLMs):** To evaluate attacks on vision-language tasks, we tested CLIP (Radford et al., 2021) for zero-shot classification, Blip2 (Li et al., 2023b) for image captioning, and LLAVA 1.5 (Liu et al., 2024) for visual question answering (VQA). In the context of Blip2, the attack is considered successful if the generated captions include the target label. Meanwhile, for LLAVA, we designed a question-prompt format, "*What is the subject in the picture?*" Here, an attack is deemed successful if the model's response contains the target label.

The results of our evaluation, presented in Table 4 and Table 5, highlight the effectiveness of resolution attacks across both labeled and unlabeled settings. For the RA attack, the low-resolution classification accuracy ($\text{Acc}_\text{L}$) remains consistently high in the unlabeled setting, achieving values such as 87.2% in ViT-l32, 91.2% in Blip2, and 90.0% in LLAVA. This is attributed to the broader semantic space of the unlabeled attack prompts, which facilitates the generation of dual-semantic images. The ability to align with both high-resolution and low-resolution categories makes the attack success rate higher in this setting. Conversely, in the labeled attack, the semantic constraints imposed by specific target labels narrow the scope of possible image variations. As a result, the challenge of generating semantically ambiguous images that satisfy both resolution levels leads to lower low-resolution classification accuracy. For the RAS attack, the introduction of source images significantly enhances control over the generated outputs, balancing high-resolution feature fidelity with low-resolution attack efficacy. In labeled attacks, where the semantic space is inherently narrower due to more specific target labels, source images guide the generation process, making it easier to create dual-semantic images. Consequently, both the low-resolution classification accuracy ($\text{Acc}_\text{L}$) and Corrective Attack Success Rate ($\text{ASR}_\text{C}$) are generally higher compared to RA. However, in the unlabeled attack, the broad semantic scope of the high-resolution prompts allows for easier genera-

Table 4: Additional quantitative results of the Resolution Attack.

| Classifiers | Labeled Attack | | | Unlabeled Attack |
|---|---|---|---|---|
| | $Acc_H \uparrow$ | $Acc_L \uparrow$ | $ASR_C \uparrow$ | $Acc_L \uparrow$ |
| ViT-b32 | 55.7% | 58.1% | 41.3% | 83.2% |
| ViT-l32 | 50.3% | 65.3% | 44.1% | 87.2% |
| Fasterrcnn_resnet50_fpn | —— | 47.7% | —— | 19.4% |
| Maskrcnn_resnet50_fpn | —— | 57.5% | —— | 29.1% |
| Clip(zero-shot) | 92.1% | 34.9% | 33.3% | 62.8% |
| Blip2(image caption) | 72.1% | 58.7% | 53.7% | 91.2% |
| LLAVA(VQA) | 71.8% | 68.9% | 63.6% | 90.0% |

Table 5: Additional quantitative results of the Resolution Attack with Source image.

| Classifiers | Labeled Attack | | | Unlabeled Attack |
|---|---|---|---|---|
| | $Acc_H \uparrow$ | $Acc_L \uparrow$ | $ASR_C \uparrow$ | $Acc_L \uparrow$ |
| ViT-b32 | 44.9% | 74.5% | 56.6% | 58.0% |
| ViT-l32 | 36.9% | 83.9% | 60.7% | 62.4% |
| Fasterrcnn_resnet50_fpn | —— | 53.5% | —— | 11.8% |
| Maskrcnn_resnet50_fpn | —— | 67.8% | —— | 18.6% |
| Clip(zero-shot) | 85.7% | 44.7% | 39.9% | 30.9% |
| Blip2(image caption) | 61.4% | 63.5% | 55.5% | 51.8% |
| LLAVA(VQA) | 63.8% | 71.6% | 59.7% | 51.4% |

tion of dual-semantic images without source image constraints. Introducing a source image in this case can limit the model's flexibility, leading to slightly lower low-resolution attack rates than RA.

These experiments collectively demonstrate the adaptability and generalizability of resolution attacks across diverse model architectures. The findings also expose significant vulnerabilities in widely-used machine learning frameworks, emphasizing the urgent need for robust defense strategies to mitigate the risks associated with resolution attacks.

## F  MORE QUALITATIVE RESULTS

We provide qualitative results in the main paper with enhanced clarity in Figure 11-13 (labeled RA), Figure 14-16 (unlabeled RA), Figure 17-19 (labeled RAS), Figure 20-22 (unlabeled RAS), Figure 33-42 (face swapping).

We also provide more qualitative results in Figure 23-24 (labeled RA), Figure 25-27 (unlabeled RA), Figure 28-29 (labeled RAS), Figure 30-32 (unlabeled RAS).

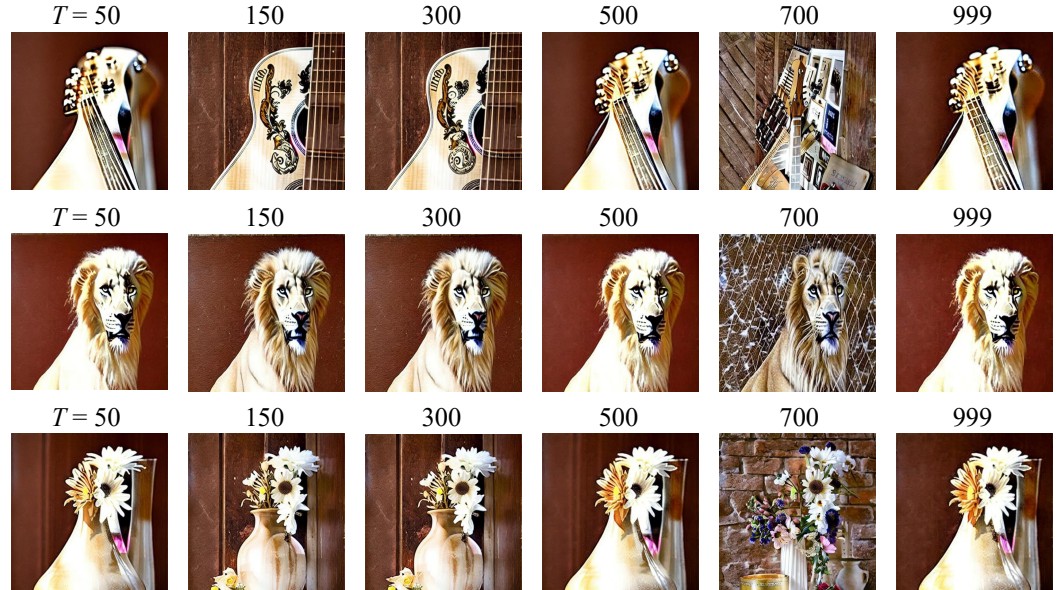

Figure 10: **Ablation on Denoising Sampling Steps.** Top: labeled object attacks. Middle: labeled animal attacks. Bottom: unlabeled attacks. $T$ represents the number of denoising sampling steps, and six different sampling steps were used: 50, 150, 300, 500, 700, and 999.

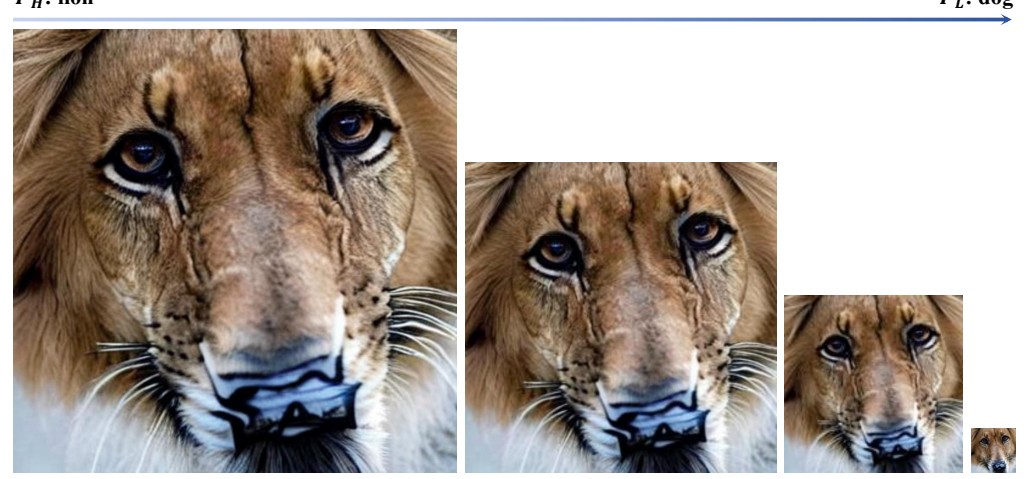

Figure 11: Qualitative results on labeled resolution attack.

$P_H$: monkey $P_L$: dog

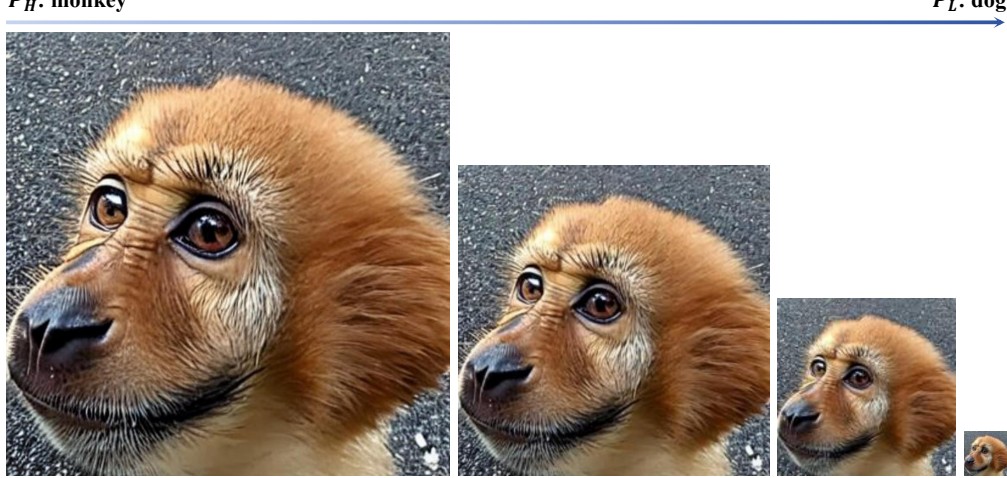

Figure 12: Qualitative results on labeled resolution attack.

$P_H$: shoe $P_L$: dog

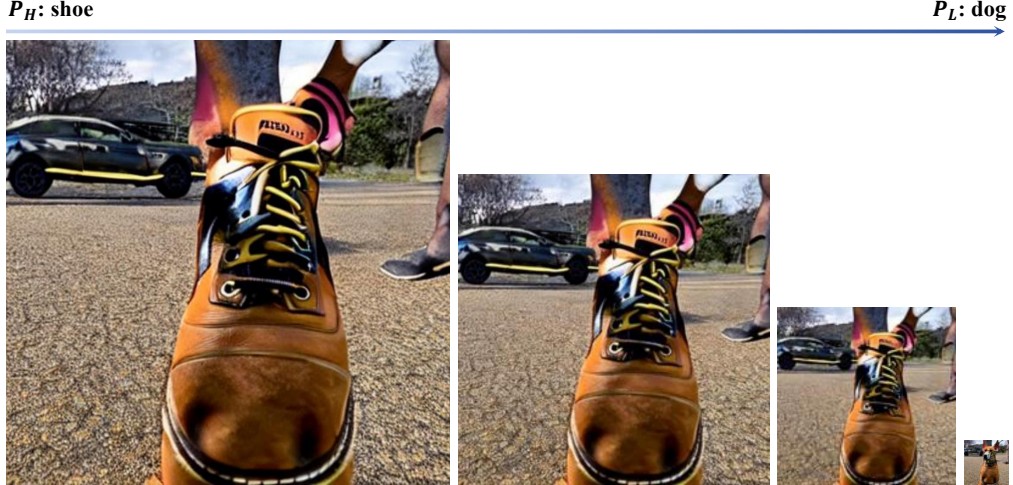

Figure 13: Qualitative results on labeled resolution attack.

$P_H$: delicious food $P_L$: dog

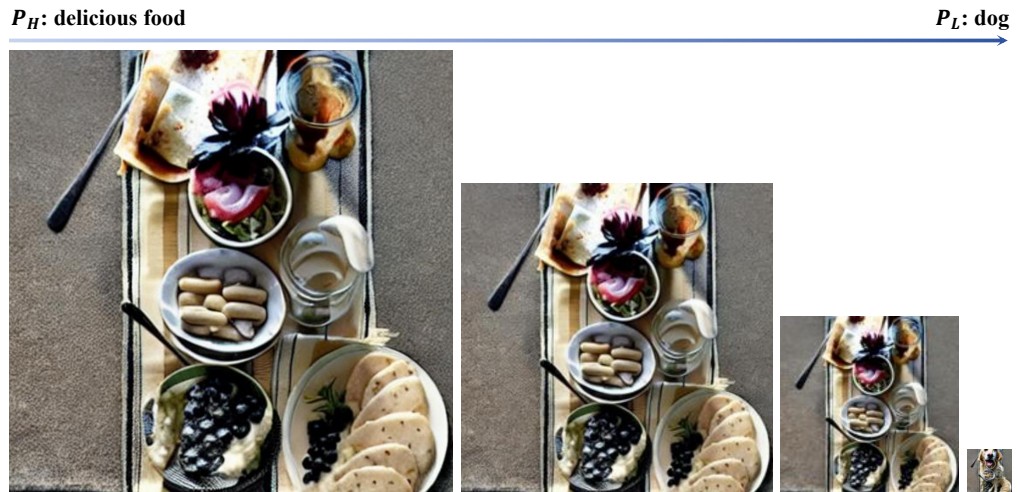

Figure 14: Qualitative results on unlabeled resolution attack.

$P_H$: a vase with flowers $\qquad\qquad\qquad\qquad\qquad\qquad\qquad\qquad$ $P_L$: dog

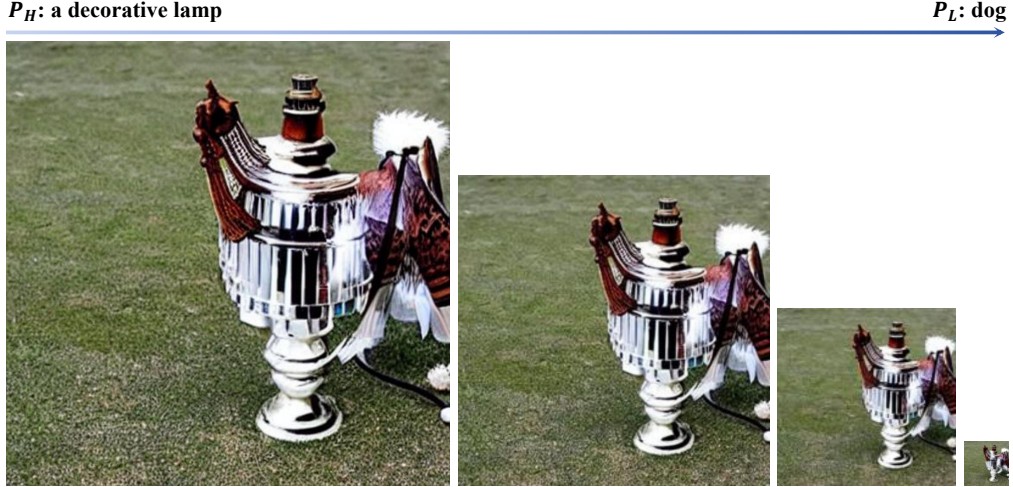

Figure 15: Qualitative results on unlabeled resolution attack.

$P_H$: a decorative lamp $\qquad\qquad\qquad\qquad\qquad\qquad\qquad\qquad$ $P_L$: dog

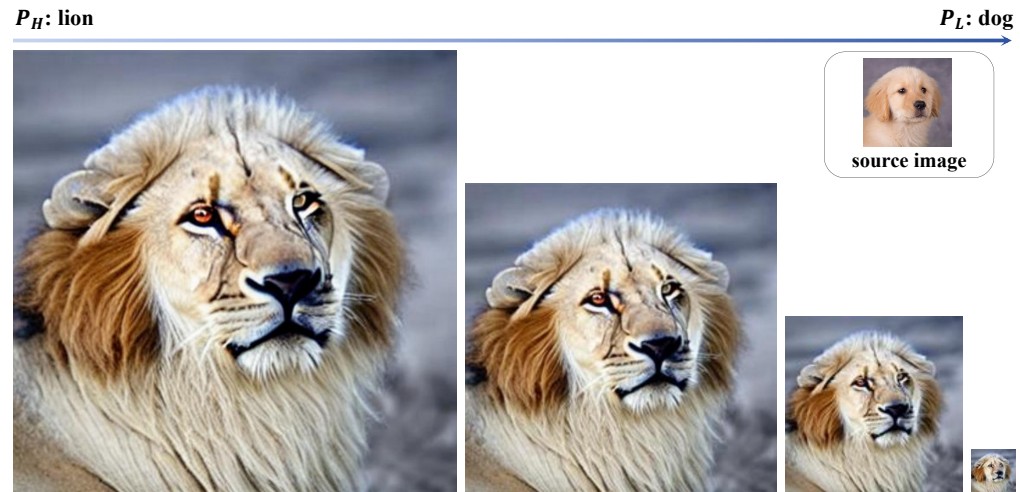

Figure 16: Qualitative results on unlabeled resolution attack.

$P_H$: lion $\qquad\qquad\qquad\qquad\qquad\qquad\qquad\qquad\qquad\qquad$ $P_L$: dog

Figure 17: Qualitative results on labeled resolution attack with source image.

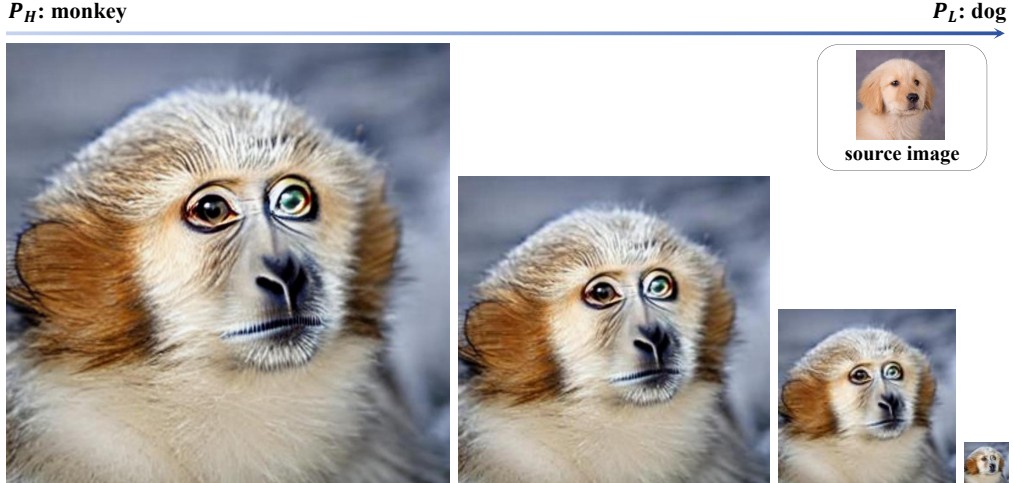

Figure 18: Qualitative results on labeled resolution attack with source image.

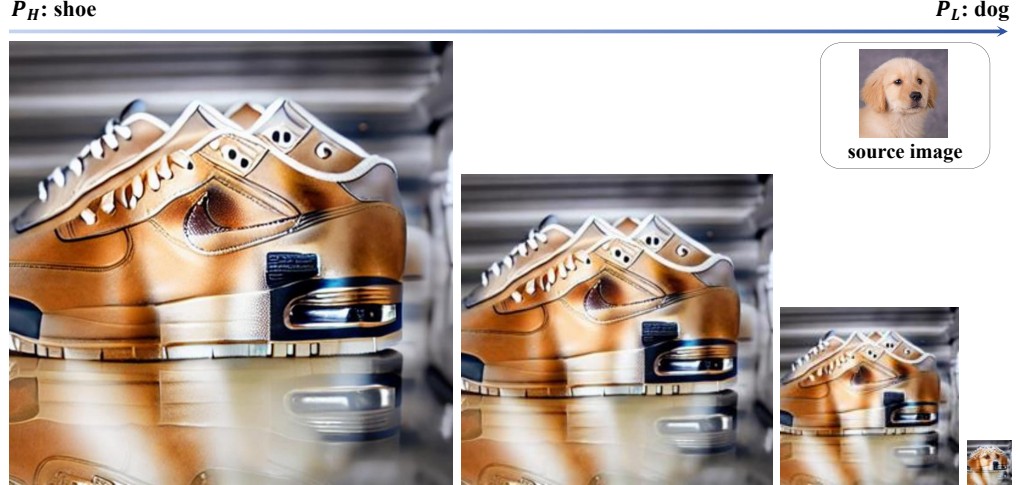

Figure 19: Qualitative results on labeled resolution attack with source image.

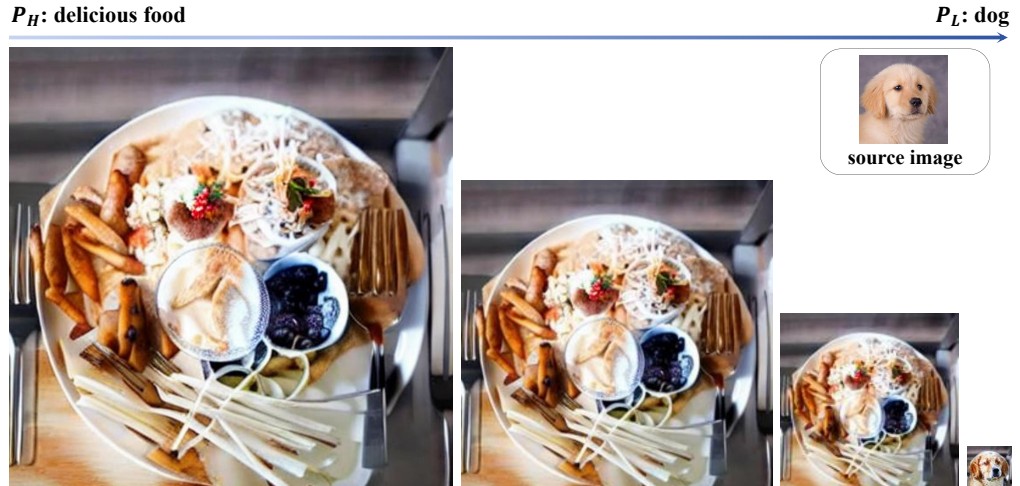

Figure 20: Qualitative results on unlabeled resolution attack with source image.

**$P_H$: a vase with flowers**                                         **$P_L$: dog**

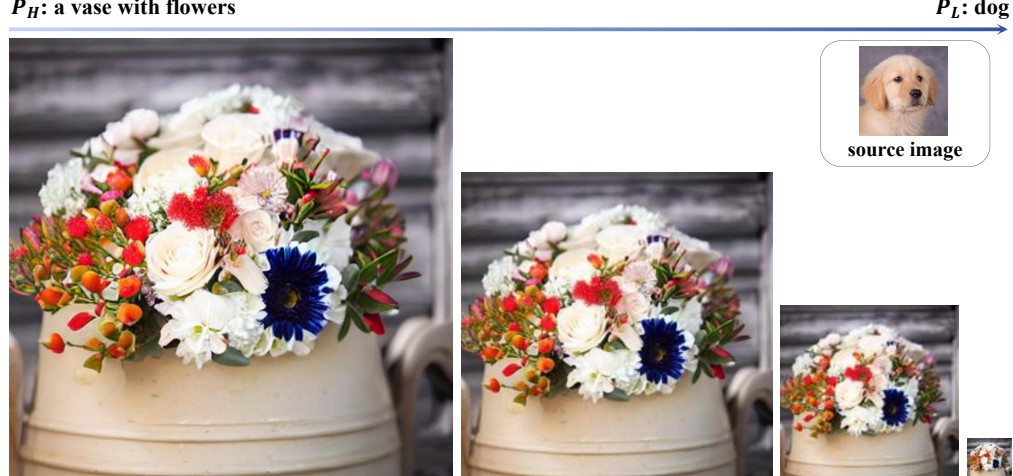

Figure 21: Qualitative results on unlabeled resolution attack with source image.

**$P_H$: a decorative lamp**                                           **$P_L$: dog**

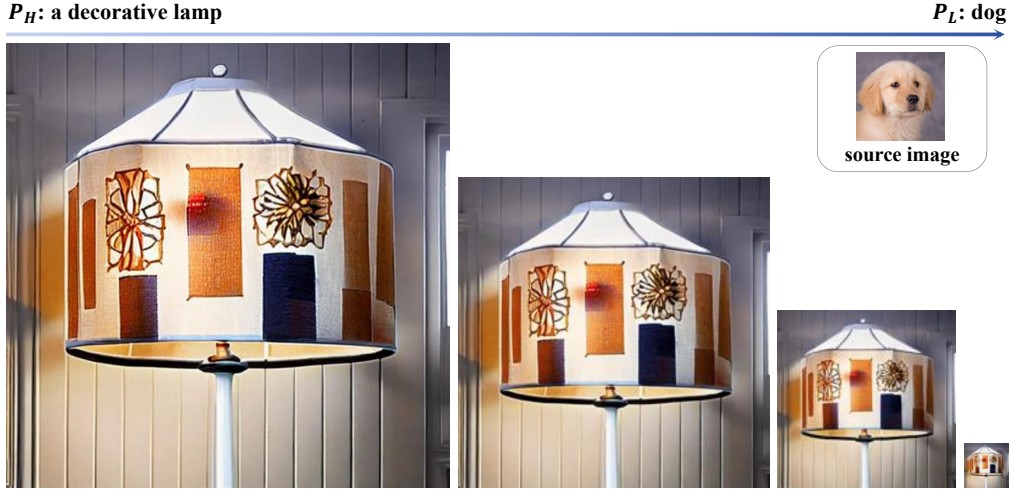

Figure 22: Qualitative results on unlabeled resolution attack with source image.

**$P_H$: guitar**                                                      **$P_L$: dog**

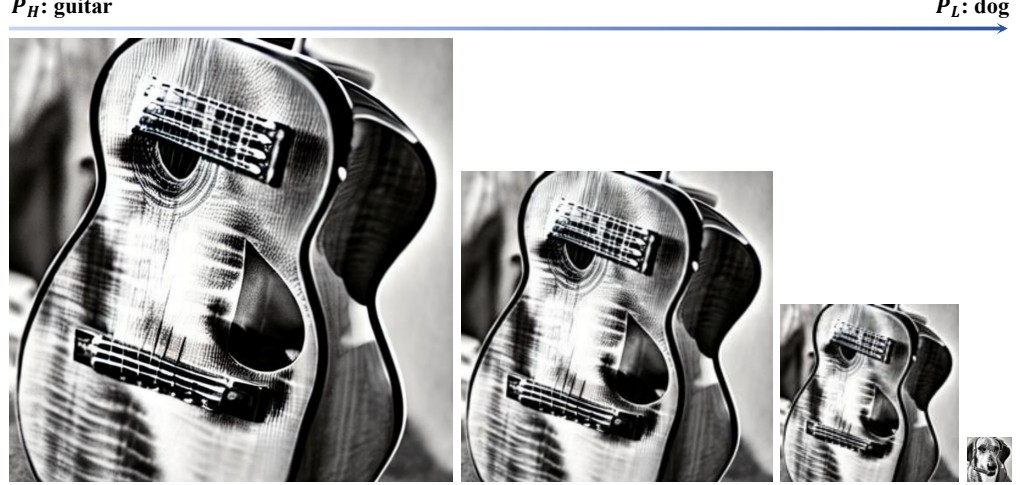

Figure 23: Qualitative results on labeled resolution attack.

$P_H$: sea lion $\qquad\qquad\qquad\qquad\qquad\qquad\qquad\qquad\qquad\qquad\qquad$ $P_L$: dog

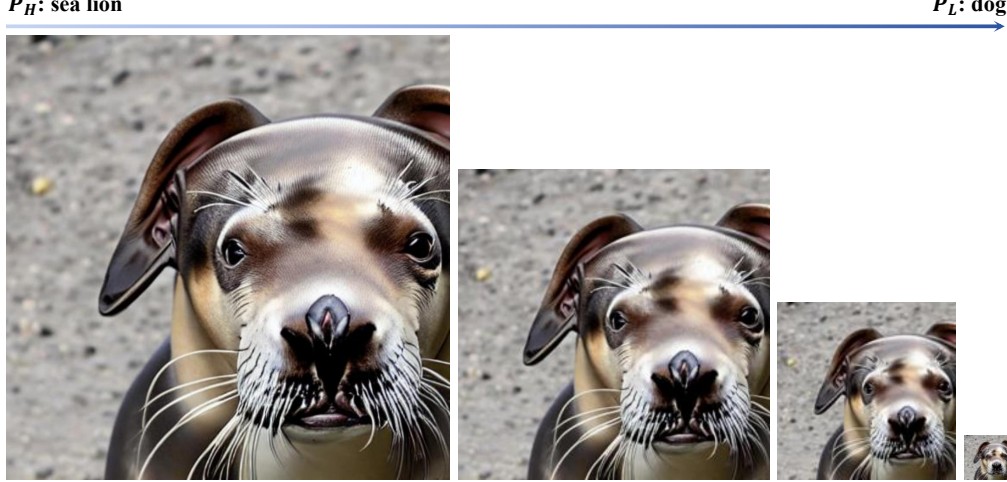

Figure 24: Qualitative results on labeled resolution attack.

$P_H$: snow village $\qquad\qquad\qquad\qquad\qquad\qquad\qquad\qquad\qquad\qquad$ $P_L$: dog

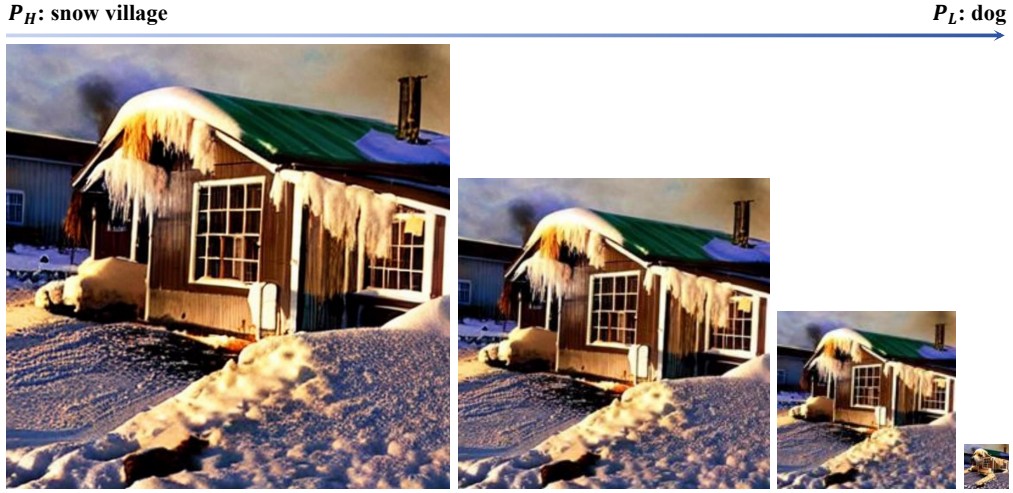

Figure 25: Qualitative results on unlabeled resolution attack.

$P_H$: calm lake $\qquad\qquad\qquad\qquad\qquad\qquad\qquad\qquad\qquad\qquad\qquad$ $P_L$: dog

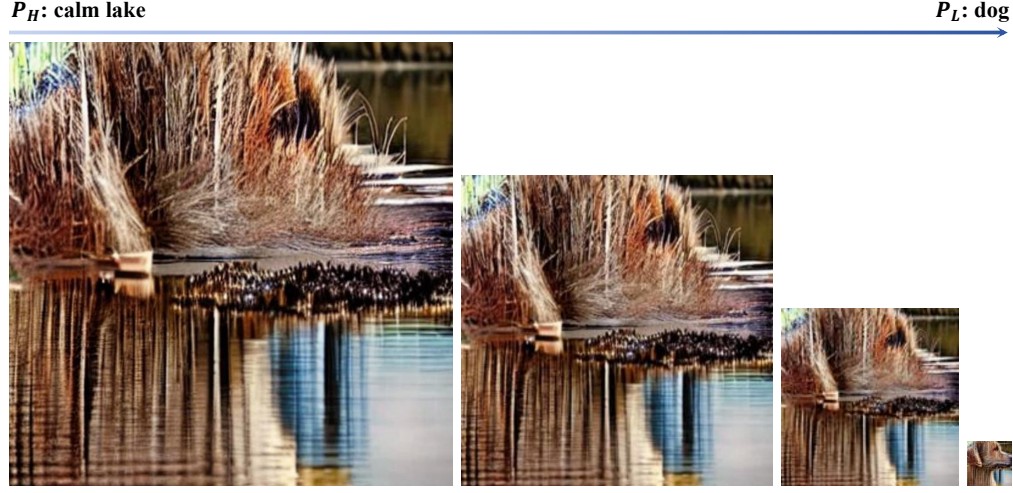

Figure 26: Qualitative results on unlabeled resolution attack.

$P_H$: mysterious cave $\qquad\qquad\qquad\qquad\qquad\qquad\qquad\qquad$ $P_L$: dog

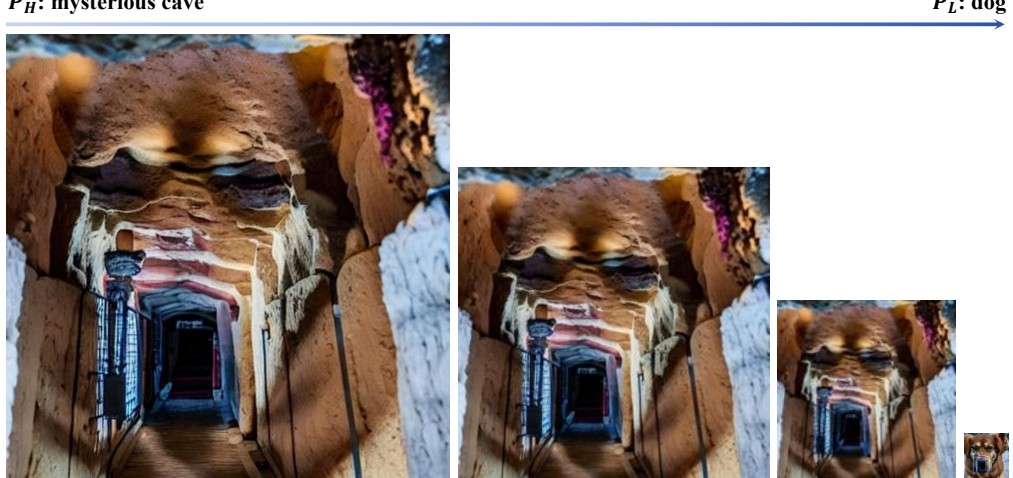

Figure 27: Qualitative results on unlabeled resolution attack.

$P_H$: guitar $\qquad\qquad\qquad\qquad\qquad\qquad\qquad\qquad\qquad\qquad$ $P_L$: dog

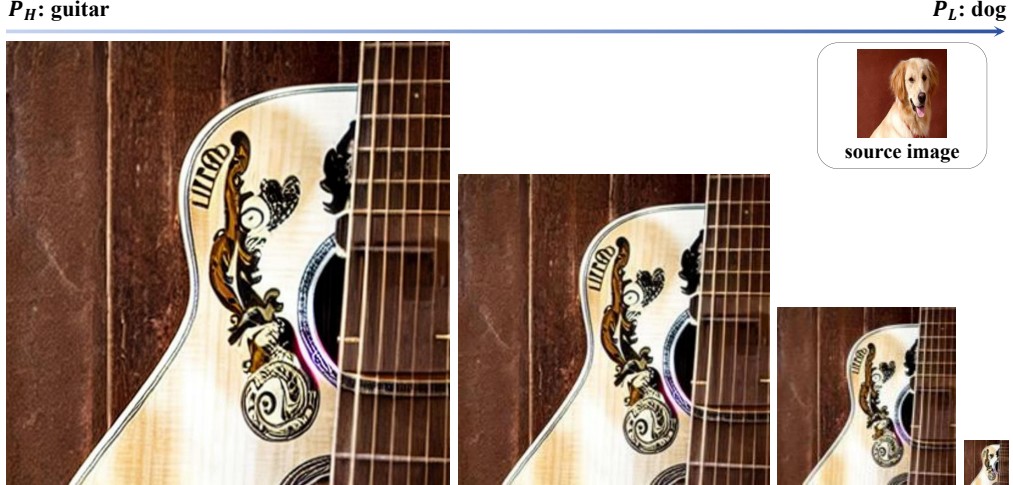

Figure 28: Qualitative results on labeled resolution attack with source image.

$P_H$: sea lion $\qquad\qquad\qquad\qquad\qquad\qquad\qquad\qquad\qquad$ $P_L$: dog

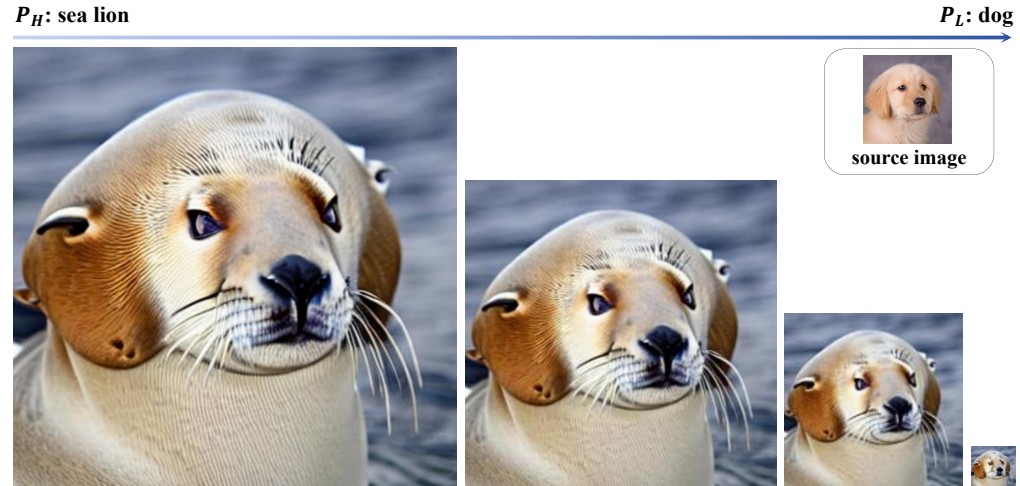

Figure 29: Qualitative results on labeled resolution attack with source image.

**$P_H$: snow village**            **$P_L$: dog**

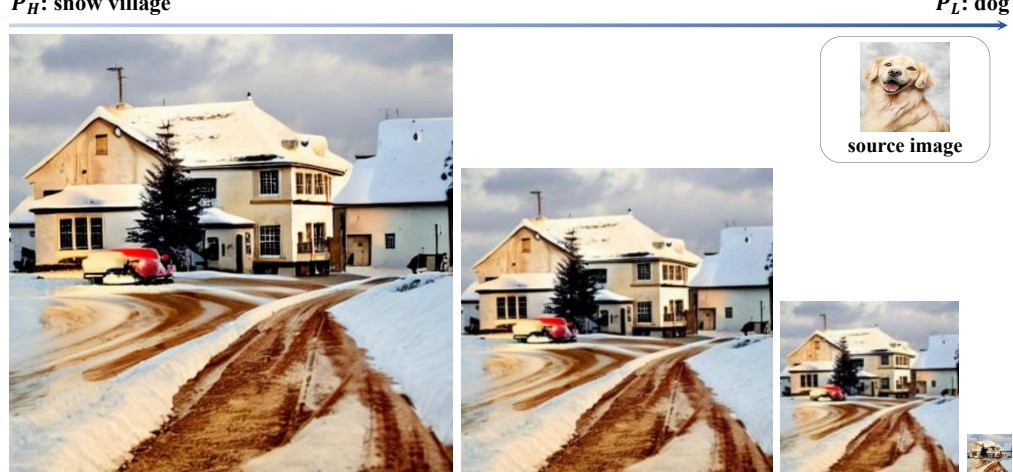

Figure 30: Qualitative results on unlabeled resolution attack with source image.

**$P_H$: calm lake**            **$P_L$: dog**

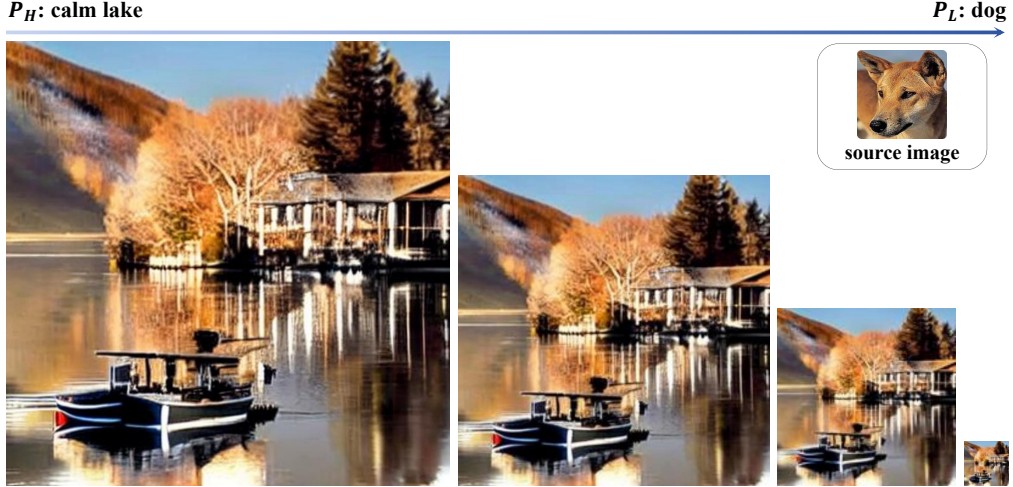

Figure 31: Qualitative results on unlabeled resolution attack with source image.

**$P_H$: mysterious cave**            **$P_L$: dog**

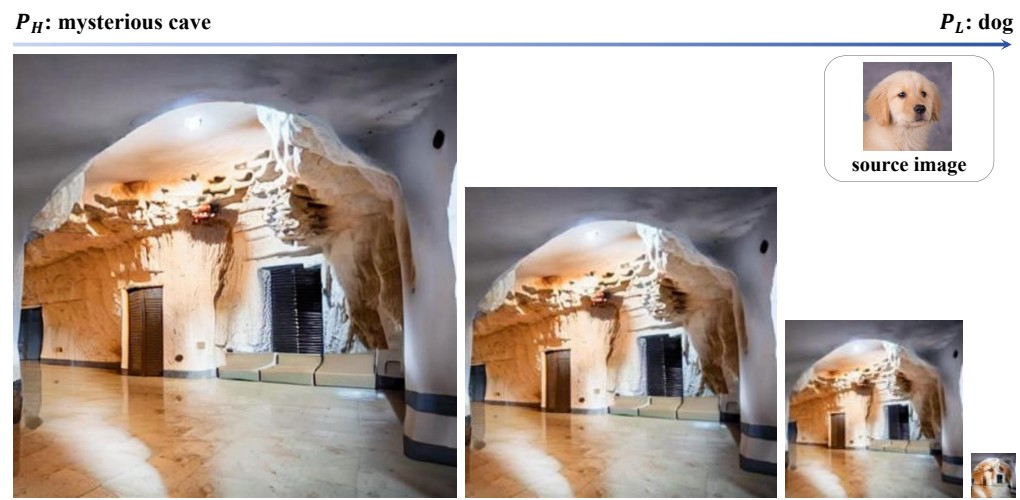

Figure 32: Qualitative results on unlabeled resolution attack with source image.

**$P_H$: Biden**                         **$P_L$: Trump**

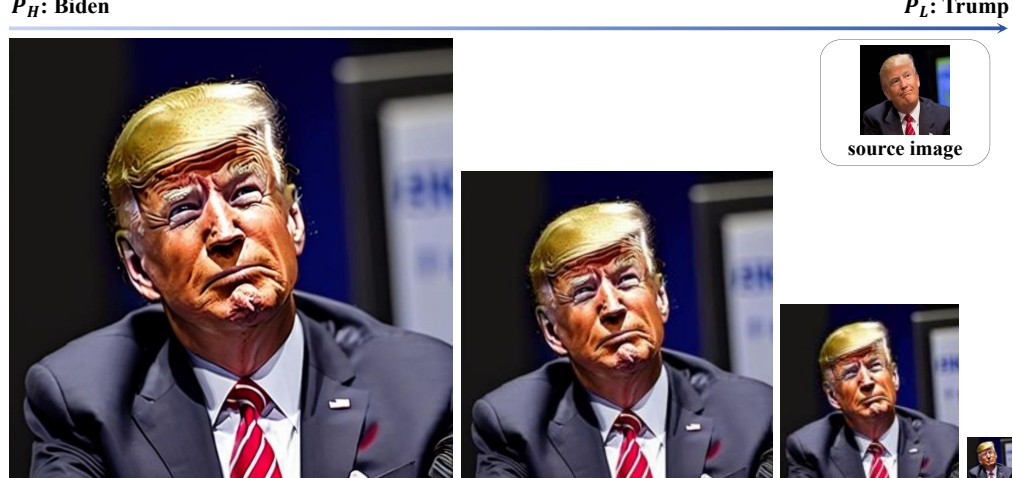

Figure 33: Qualitative results of face swapping.

**$P_H$: Boris Johnson**                     **$P_L$: Trump**

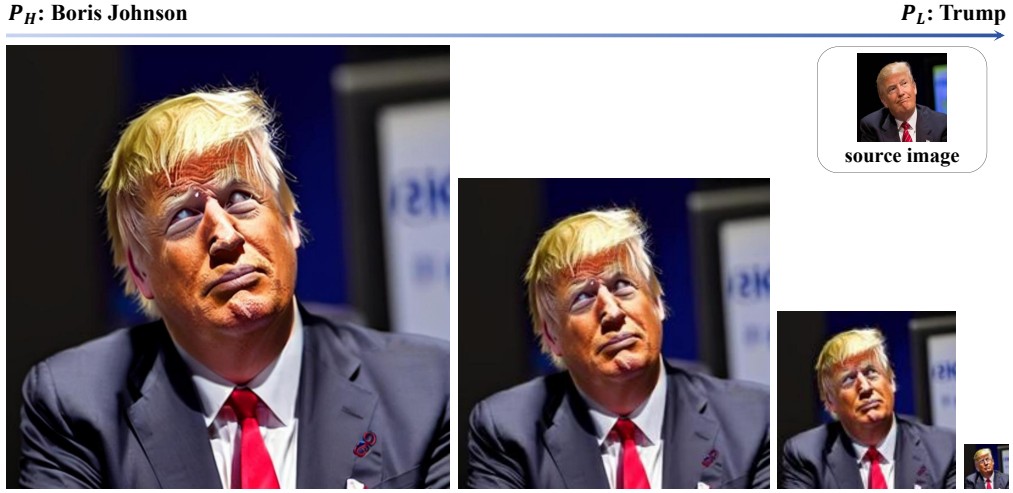

Figure 34: Qualitative results of face swapping.

**$P_H$: Bill Gates**                      **$P_L$: Trump**

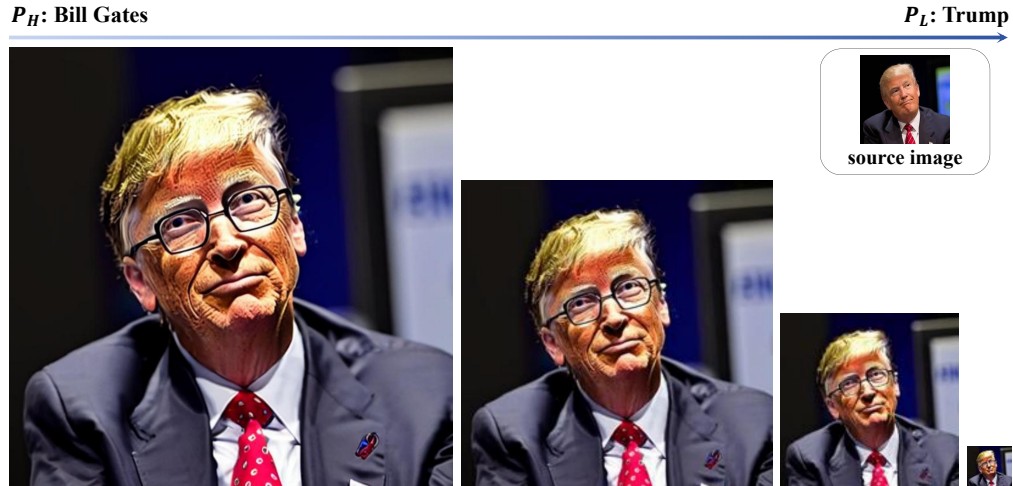

Figure 35: Qualitative results of face swapping.

$P_H$: Mark Zuckerberg $\qquad\qquad\qquad\qquad\qquad\qquad\qquad\qquad$ $P_L$: Trump

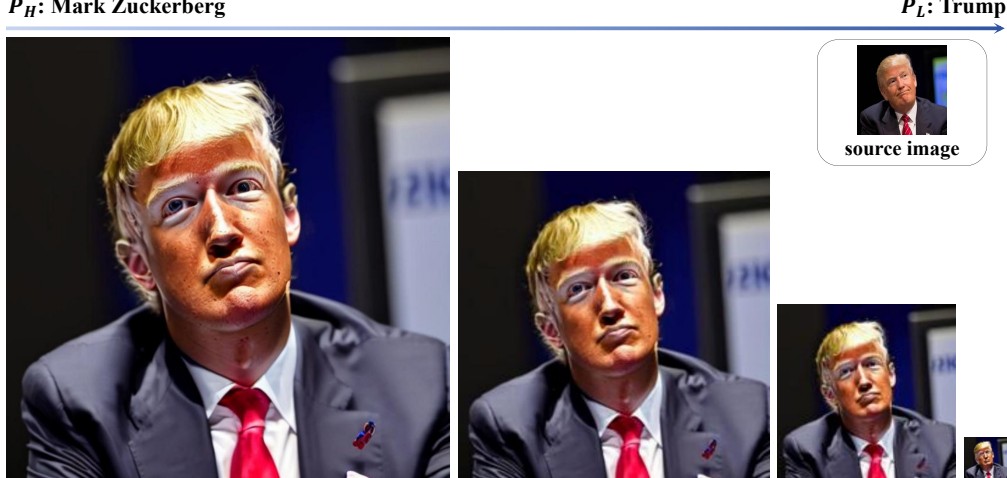

Figure 36: Qualitative results of face swapping.

$P_H$: Steve Jobs $\qquad\qquad\qquad\qquad\qquad\qquad\qquad\qquad\qquad$ $P_L$: Trump

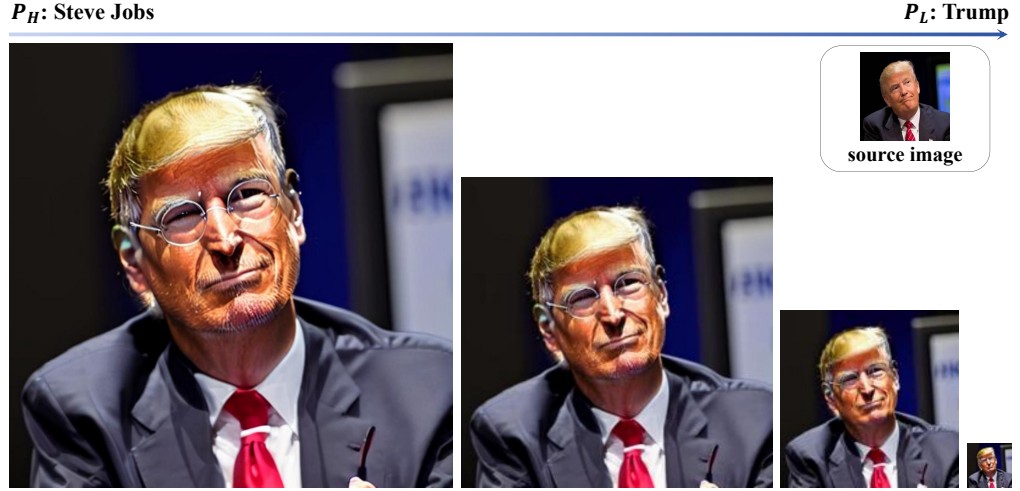

Figure 37: Qualitative results of face swapping.

$P_H$: Biden $\qquad\qquad\qquad\qquad\qquad\qquad\qquad\qquad\qquad\qquad$ $P_L$: Musk

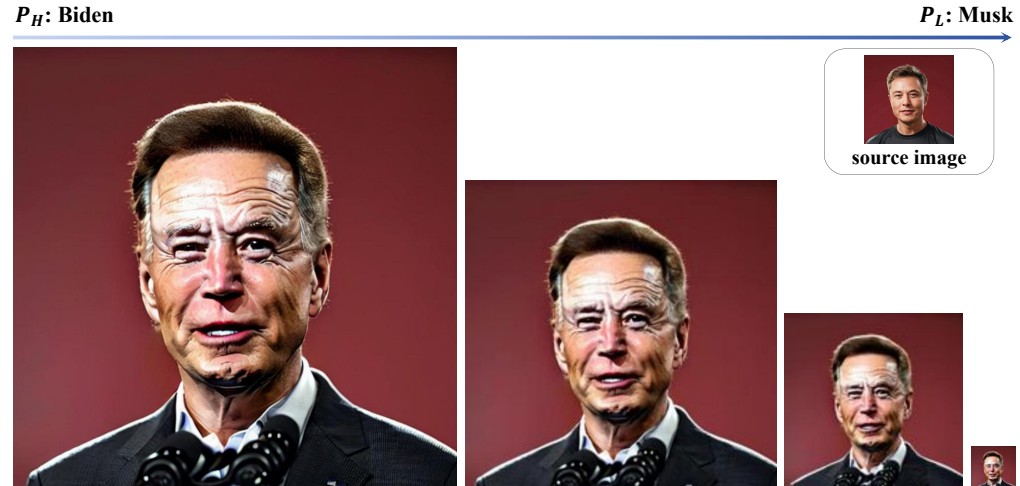

Figure 38: Qualitative results of face swapping.

**$P_H$: Boris Johnson**                                    **$P_L$: Musk**

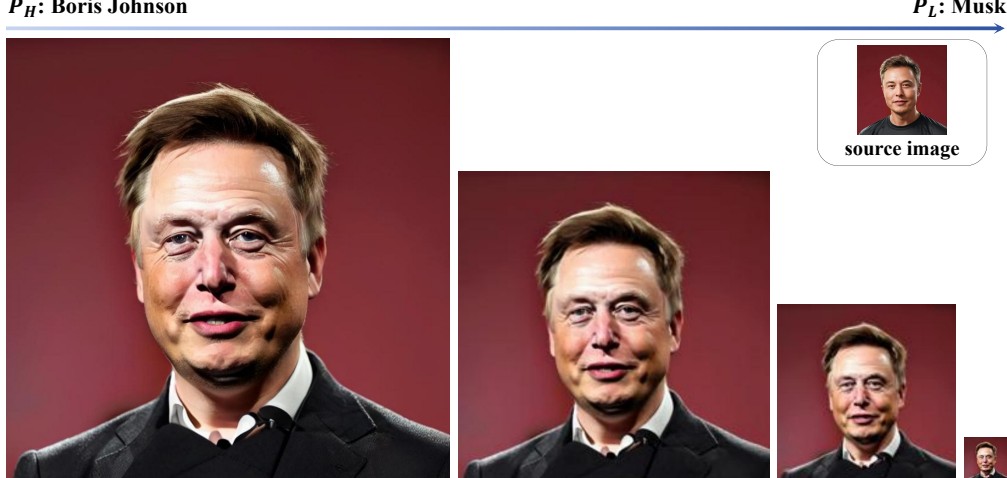

Figure 39: Qualitative results of face swapping.

**$P_H$: Bill Gates**                                       **$P_L$: Musk**

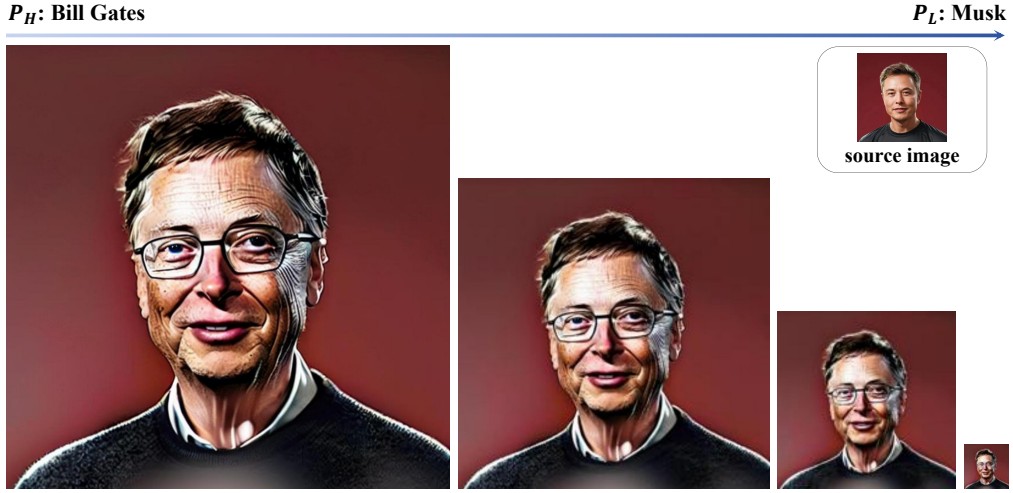

Figure 40: Qualitative results of face swapping.

**$P_H$: Mark Zuckerberg**                                  **$P_L$: Musk**

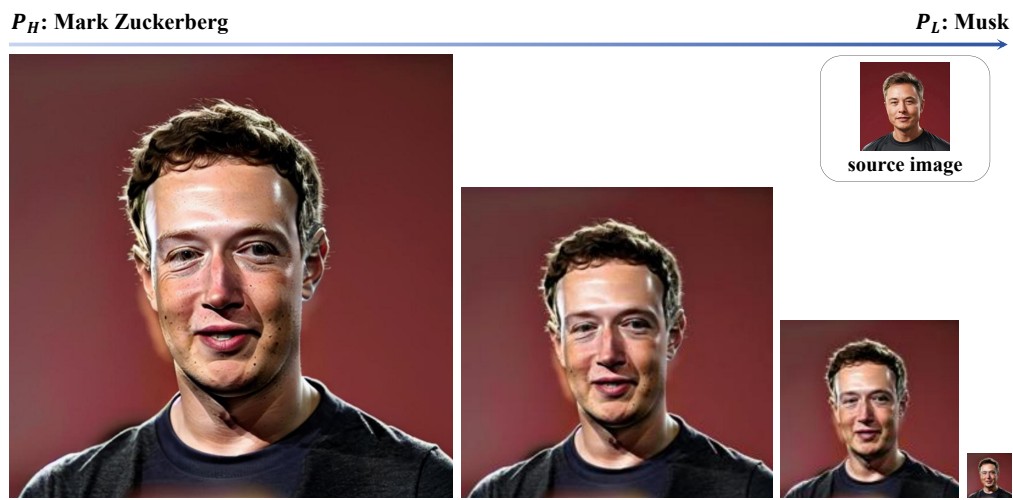

Figure 41: Qualitative results of face swapping.

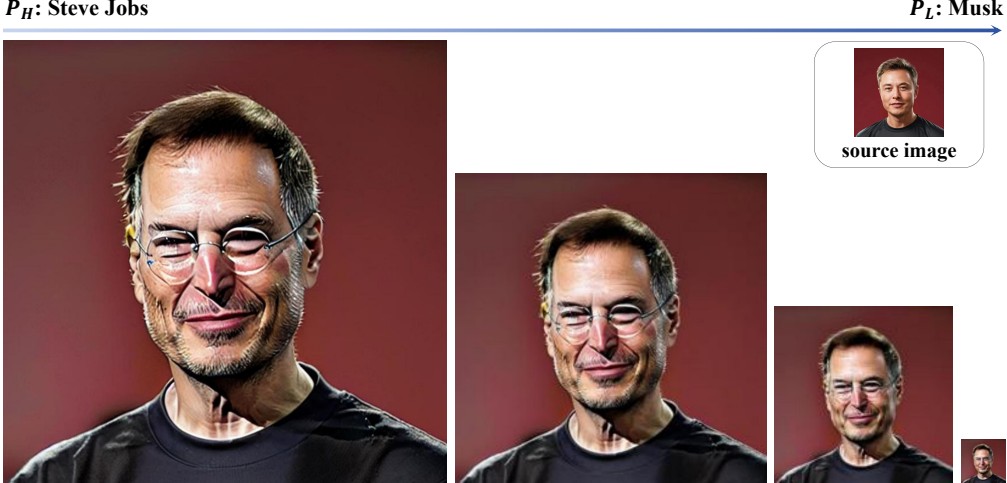

$P_H$: Steve Jobs                                                                                        $P_L$: Musk

source image

Figure 42: Qualitative results of face swapping.

