# OpenReview forum: "Resolution Attack: Exploiting Image Compression to Deceive Deep Neural Networks"
_ICLR.cc/2025/Conference — ICLR 2025 Poster_

### Official Review · Reviewer_6DtR · 2024-10-29

**Soundness:** 4
**Presentation:** 4
**Contribution:** 3
**Rating:** 6
**Confidence:** 5

**Summary:**

This paper introduces the “Resolution Attack”, a novel form of adversarial attack. Specifically, it generates images with semantic ambiguity across different resolutions to achieve the attack. To achieve this, the paper leverages diffusion models and a proposed staged denoising strategy to introduce two types of resolution attacks: RA (adversarial samples generated conditioned on dual text) and RAS (adversarial samples generated conditioned on images). Experimental results demonstrate a high attack success rate and show further potential in applications such as facial forgery.

**Strengths:**

- Well, I feel this work interesting as it presents an attack manner that is fundamentally different from previous adversarial attack approaches, that, leveraging the dual semantics introduced by resolution changes to confuse classifiers, which is quite refreshing.
- The authors' proposed optimization framework incorporates a Dual-Stream Denoising (DS Module) that locates perturbation regions in high- and low-frequency areas to achieve the attack, providing a good baseline for this attack manner.
- The experiments and ablation studies presented are comprehensive and solid.

**Weaknesses:**

Some concerns:
- (minor, discussion) Does the author thoroughly investigate the problem being studied? Specifically, are there existing resolution-based attack methods, and if so, should they be considered baselines? I suggest the author provide a more in-depth discussion of related work.
- (major, experimental) Although the effectiveness of the attack method on deep classifiers has been well demonstrated, all the models used are CNN-based classifiers with relatively low input resolution (224×224). In contrast, I am more interested in whether these adversarial samples can disrupt existing vision-language models (VLMs, e.g., CLIP, BLIP-2, LLAVA, etc.), which possess more excellent generalization capabilities than task-specific models. I think adding some evaluation results on VLMs, such as zero-shot classification or even image captioning and VQA, would significantly enhance the assessment of this method's value.
- (minor, experimental) I suggest including experiments under defensive settings to evaluate whether these adversarial samples can still successfully confuse classifiers when facing adversarial preprocessing (e.g., different resolution compression methods or adversarial purification).
- (minor discussion) Given the potential risks associated with this work, I also recommend discussing the societal impacts and potential defence strategies at the end of the paper.

**Questions:**

This paper is sufficient excellent, but I hope the authors can address the above concerns.

**Details Of Ethics Concerns:**

This paper proposes a novel adversarial attack method that may prevent harmful semantic content from being effectively detected by current technologies.

---

> ### Author Response · Authors · 2024-11-24
> **Response to Reviewer 6DtR (1/3)**
>
> We express our gratitude to the reviewers for their meticulous assessment and valuable suggestions. Your detailed feedback is paramount in refining our work, and we have undertaken measures to address the concerns.
>
> > Weakness 1: (minor, discussion) Does the author thoroughly investigate the problem being studied? Specifically, are there existing resolution-based attack methods, and if so, should they be considered baselines? I suggest the author provide a more in-depth discussion of related work.
>
> We appreciate the reviewer’s insightful comment on the discussion of related work. **To our best knowledge, we are the first to propose the Resolution Attack (RA), an attack that is different from current arts.** In the revised paper (Appendix D), we have elaborated on existing generative adversarial approaches in latent space, such as those leveraging diffusion models to manipulate U-Net parameters or introduce learned perturbations in latent space. While these works focus on adversarial attacks in generative modeling, **none explore the unique dual-representation mechanism proposed in our resolution attacks.** By introducing resolution attacks, we aim to provide a foundational framework for future research in this direction, fostering deeper understanding and advancements in the field.

---

> ### Author Response · Authors · 2024-11-24
> **Response to Reviewer 6DtR (2/3)**
>
> > Weakness 2: (major, experimental) Although the effectiveness of the attack method on deep classifiers has been well demonstrated, all the models used are CNN-based classifiers with relatively low input resolution (224×224). In contrast, I am more interested in whether these adversarial samples can disrupt existing vision-language models (VLMs, e.g., CLIP, BLIP-2, LLAVA, etc.), which possess more excellent generalization capabilities than task-specific models. I think adding some evaluation results on VLMs, such as zero-shot classification or even image captioning and VQA, would significantly enhance the assessment of this method's value.
>
> Thanks for your valuable insights. **As you have pointed out, our method focus on the semantic of the images and is both model-agnostic and task-agnostic.** To demonstrate this, we conduct experiments across various architectures (ViTs, FPN) and various tasks (VQA, Image Caption, Zero-Shot Classification). The results are shown in the following table:
>
>      Additional quantitative results of the Resolution Attack
> |        Classifiers        |                          |      Labeled Attack      |                          |   |   |     Unlabeled Attack     |
> |:-------------------------:|:------------------------:|:------------------------:|:------------------------:|---|---|:------------------------:|
> |                           | $\mathrm{Acc_H}\uparrow$ | $\mathrm{Acc_L}\uparrow$ | $\mathrm{ASR_C}\uparrow$ |   |   | $\mathrm{Acc_L}\uparrow$ |
> |          ViT-b32          |          55.7\%          |          58.1\%          |          41.3\%          |   |   |          83.2\%          |
> |          ViT-l32          |          50.3\%          |          65.3\%          |          44.1\%          |   |   |          87.2\%          |
> | Fasterrcnn\_resnet50\_fpn |            ——            |          47.7\%          |            ——            |   |   |          19.4\%          |
> |  Maskrcnn\_resnet50\_fpn  |            ——            |          57.5\%          |            ——            |   |   |          29.1\%          |
> |      Clip(zero-shot)      |          92.1\%          |          34.9\%          |          33.3\%          |   |   |          62.8\%          |
> |    Blip2(image caption)   |          72.1\%          |          58.7\%          |          53.7\%          |   |   |          91.2\%          |
> |         LLAVA(VQA)        |          71.8\%          |          68.9\%          |          63.6\%          |   |   |          90.0\%          |
>
>      Additional quantitative results of the Resolution Attack with Source image
> |        Classifiers        |                          |      Labeled Attack      |                          |   |   |     Unlabeled Attack     |
> |:-------------------------:|:------------------------:|:------------------------:|:------------------------:|---|---|:------------------------:|
> |                           | $\mathrm{Acc_H}\uparrow$ | $\mathrm{Acc_L}\uparrow$ | $\mathrm{ASR_C}\uparrow$ |   |   | $\mathrm{Acc_L}\uparrow$ |
> |          ViT-b32          |          44.9\%          |          74.5\%          |          56.6\%          |   |   |          58.0\%          |
> |          ViT-l32          |          36.9\%          |          83.9\%          |          60.7\%          |   |   |          62.4\%          |
> | Fasterrcnn\_resnet50\_fpn |            ——            |          53.5\%          |            ——            |   |   |          11.8\%          |
> |  Maskrcnn\_resnet50\_fpn  |            ——            |          67.8\%          |            ——            |   |   |          18.6\%          |
> |      Clip(zero-shot)      |          85.7\%          |          44.7\%          |          39.9\%          |   |   |          30.9\%          |
> |    Blip2(image caption)   |          61.4\%          |          63.5\%          |          55.5\%          |   |   |          51.8\%          |
> |         LLAVA(VQA)        |          63.8\%          |          71.6\%          |          59.7\%          |   |   |          51.4\%          |
>
> The results demonstrate that our RA attack achieve over 90% success rates in unlabeled attacks on Blip2 and LLAVA 1.5. Similarly, in RAS attack, these models also exhibit strong attack susceptibility. The results demonstrate that our generated images can effectively attack various models across various vision tasks. We have incorporated these experiments in Appendix E of in the revised paper.

---

> ### Author Response · Authors · 2024-11-24
> **Response to Reviewer 6DtR (3/3)**
>
> > Weakness 3: (minor, experimental) I suggest including experiments under defensive settings to evaluate whether these adversarial samples can still successfully confuse classifiers when facing adversarial preprocessing (e.g., different resolution compression methods or adversarial purification).
>
> We thank the reviewer for the suggestion. Specifically, we tested our RAS attack with additional compression techniques, including bilinear interpolation (INTER_LINEAR) and antialiasing (ANTIALIAS), alongside the default bicubic interpolation. Results are as follows:
>
> |              |  $\mathrm{Acc_L}\uparrow$ |
> |:------------:|:-----:|
> | INTER_LINEAR | 71.2% |
> |    BICUBIC   | 71.8% |
> |   ANTIALIAS  | 69.8% |
>
> These results indicate that the choice of compression method has minimal impact on attack success rates, because our attacks are semantics-driven and less sensitive to pixel-level transformations.
>
> We apologize for not conducting adversarial purification experiments due to time constraints. However, we are optimistic about the experimental results: current purification methods, targeted for traditional pixel-level attacks, typically embed purification into the denoising process, neutralizing adversarial perturbations and Gaussian noise simultaneously. While our method compose semantic-level manipulation which is out of the scope of current purification methods. We will explore it as well as the effective defense method specially tailored for our proposed Resolution Attack for future work.
>
> > Weakness 4: (minor discussion) Given the potential risks associated with this work, I also recommend discussing the societal impacts and potential defence strategies at the end of the paper.
>
> We appreciate the reviewer’s recommendation to discuss societal implications and defense strategies. In the revised paper (Appendix G), we have added a detailed discussion. Our method is designed to explore vulnerabilities in machine learning classifiers and assess robustness by generating dual-representation images. Though our approach is under the risk of malicious purpose, our generated image can be employed to fine-tune existing classifiers, further improving their robustness. Besides, the generated images can also be detected by deepfake detectors.

---

> ### Comment · Reviewer_6DtR · 2024-11-28
>
> Thanks for your response. My concerns have been addressed so I decide to raise my confidence score.

---

> > ### Author Response · Authors · 2024-11-28
> > **Thank you for your recognition of our work**
> >
> > We would like to sincerely thank you for your professional review and constructive feedback on our paper. We are especially grateful for your recognition, describing our work as “sufficient excellent,” which is highly encouraging and motivates us to continue refining our research.
> >
> > Your suggestions have significantly contributed to the improvement of our paper. In particular, your recommendation to evaluate our method on vision-language models (VLMs), such as CLIP, BLIP-2, and LLAVA, has greatly enhanced the generalization aspect of our work. This feedback helped us demonstrate the broader applicability of our method across diverse model architectures, and we appreciate the opportunity to include these evaluations in our research. Additionally, your other comments, such as discussing societal impacts and defense strategies, have deepened the scope of our work and made it more comprehensive.
> >
> > Once again, we are deeply grateful for your recognition of the quality of our work and for your valuable suggestions, which have helped us further refine and improve our paper. Thank you for your time and effort in reviewing our research and for your thoughtful contributions to its development.

---

### Official Review · Reviewer_XjQe · 2024-11-03

**Soundness:** 3
**Presentation:** 3
**Contribution:** 3
**Rating:** 6
**Confidence:** 2

**Summary:**

In this paper, the authors propose a new attack method called as resolution attack, i.e., an automated low-resolution image generative framework capable of generating dual-semantic images in a zero-shot manner. Experiments are conducted to verify the effectiveness of resolution attack.

**Strengths:**

1. A new method called as resolution attack is proposed, which is interesting.
2.  A reasonable technique is proposed to implement the proposed idea
3. The paper is easily understood.

**Weaknesses:**

1. It seems the test images are important for the success of resolution attack, because when an image is compressed into a low-resolution image, the semantic content is easily changed. In the paper, the authors don’t give the test dataset, so I don’t know whether the proposed method  is still effective for all images.
2. The method should be compared with the compression method that randomly reduce the resolution of the test image. Compared with this baseline, the readers can better see the effectiveness of this attack.
3. Could the proposed method is till effective for vision transformer? The authors should give the discussion.

**Questions:**

see the weakness

---

> ### Author Response · Authors · 2024-11-24
> **Response to Reviewer XjQe (1/2)**
>
> We sincerely appreciate the insightful feedback from the reviewers, and we have diligently addressed the raised concerns to enhance the quality of our work.
>
> >Weakness 1: It seems the test images are important for the success of resolution attack, because when an image is compressed into a low-resolution image, the semantic content is easily changed. In the paper, the authors don’t give the test dataset, so I don’t know whether the proposed method is still effective for all images.
>
> Thanks for this advice. We would like to note that **the Resolution Attack, which is included in the proposed settings, generates images from scratch without the requirement of a clean test image. Here, we mainly want to detail the setting that require clean test images (Resolution Attack with Source image, RAS).** As we have stated in the original submission (Lines 415-416), we collect 100 frontal images as the source images due to the reason that facial features are critical for semantic representation. Similarly, in the following experiments on human faces (“Resolution Attcaks as the Face Swapper”), we also adopt human face images with clear frontal face.
>
> To further demonstrate that our method is not sensitive to the choice of the semantic of the source images, **we evaluate our method in broader classes**. Specifically, we leverage 10 additional classes including boat, bird, cat et.al. which are in the class set of Cifar10 dataset. We exclude the class ‘deer’ and ‘horse’ as they are not in the label sets of target classifiers. The results are presented below:
>
> |                           | boat | bird | car | cat | dog | frog | plane | truck |
> |:-------------------------:|:----:|:----:|:---:|:---:|:---:|:----:|:-----:|:-----:|
> |  Resnet50-Labeled Attack  |  84% |  38% | 80% | 27% | 67% |  17% |  26%  |  28%  |
> |   Vit-l32-Labeled Attack  |  79% |  43% | 75% | 62% | 74% |  61% |  79%  |  75%  |
> | Resnet50-Unlabeled Attack |  68% |  54% | 55% | 23% | 60% |  16% |  20%  |  36%  |
> |  Vit-l32-Unlabeled Attack |  74% |  73% | 76% | 63% | 88% |  72% |  80%  |  84%  |
>
> These results demonstrate that **our proposed method achieves robust attack performance across most categories, highlighting its generalizability.**
>
> > Weakness 2: The method should be compared with the compression method that randomly reduce the resolution of the test image. Compared with this baseline, the readers can better see the effectiveness of this attack.
>
> **We first clarify that while directly applying compression operation can cause misclassifications, they cannot achieve targeted labels, unlike our proposed Resolution Attack, which achieves targeted classifications.**
>
> Moreover, current classifiers exhibit a certain degree of robustness to compression. To demonstrate this, we conduct experiments to evaluate the accuracy of current classifiers for both compressed natural images and generated images which is generated by SD.
>
>      Accuracy of Random Image Compression
> | Classifiers | Compressed Natural Images | Compressed Generated Images |
> |:-----------:|:-------------------------:|:---------------------------:|
> |   Resnet50  |            76%            |             55%             |
> |    Vgg19    |            72%            |             67%             |
> |   Vit-l32   |             97%            |              96%             |
> |   Vit-b32   |             94%            |              94%             |
>
> For traditional convolutional networks (ResNet50, VGG19), compression led to a higher proportion of misclassified images. Notably, classes such as "otter," "bear," "shoe," and "guitar" were more prone to errors, **often being misclassified into similar or adjacent categories (e.g., "otter" being classified as "weasel" or "marmot")**. In contrast, vision transformers (e.g., ViT-L32, ViT-B32) were significantly more robust to compression, resulting in fewer misclassified images.

---

> ### Author Response · Authors · 2024-11-24
> **Response to Reviewer XjQe (2/2)**
>
> > Weakness 3: Could the proposed method is till effective for vision transformer? The authors should give the discussion.
>
> Appreciate the insightful query. We have conducted our proposed Resolution Attacks on Transformer-based models (ViT-B and ViT-L). Besides, according to suggestion provided by Reviewer dpTx and 6DtR, we have evaluated the effectiveness on Feature Pyramid Network (FPN) methods (Faster RCNN and Mask RCNN) and various vision-language models, including image captioning models (BLIP), Visual Question Answering models (LLAVA) and zero-shot classification models (CLIP). The results are shown in the following tables:
>
>      Additional quantitative results of the Resolution Attack
> |        Classifiers        |                          |      Labeled Attack      |                          |   |   |     Unlabeled Attack     |
> |:-------------------------:|:------------------------:|:------------------------:|:------------------------:|---|---|:------------------------:|
> |                           | $\mathrm{Acc_H}\uparrow$ | $\mathrm{Acc_L}\uparrow$ | $\mathrm{ASR_C}\uparrow$ |   |   | $\mathrm{Acc_L}\uparrow$ |
> |          ViT-b32          |          55.7\%          |          58.1\%          |          41.3\%          |   |   |          83.2\%          |
> |          ViT-l32          |          50.3\%          |          65.3\%          |          44.1\%          |   |   |          87.2\%          |
> | Fasterrcnn\_resnet50\_fpn |            ——            |          47.7\%          |            ——            |   |   |          19.4\%          |
> |  Maskrcnn\_resnet50\_fpn  |            ——            |          57.5\%          |            ——            |   |   |          29.1\%          |
> |      Clip(zero-shot)      |          92.1\%          |          34.9\%          |          33.3\%          |   |   |          62.8\%          |
> |    Blip2(image caption)   |          72.1\%          |          58.7\%          |          53.7\%          |   |   |          91.2\%          |
> |         LLAVA(VQA)        |          71.8\%          |          68.9\%          |          63.6\%          |   |   |          90.0\%          |
>
>      Additional quantitative results of the Resolution Attack with Source image
> |        Classifiers        |                          |      Labeled Attack      |                          |   |   |     Unlabeled Attack     |
> |:-------------------------:|:------------------------:|:------------------------:|:------------------------:|---|---|:------------------------:|
> |                           | $\mathrm{Acc_H}\uparrow$ | $\mathrm{Acc_L}\uparrow$ | $\mathrm{ASR_C}\uparrow$ |   |   | $\mathrm{Acc_L}\uparrow$ |
> |          ViT-b32          |          44.9\%          |          74.5\%          |          56.6\%          |   |   |          58.0\%          |
> |          ViT-l32          |          36.9\%          |          83.9\%          |          60.7\%          |   |   |          62.4\%          |
> | Fasterrcnn\_resnet50\_fpn |            ——            |          53.5\%          |            ——            |   |   |          11.8\%          |
> |  Maskrcnn\_resnet50\_fpn  |            ——            |          67.8\%          |            ——            |   |   |          18.6\%          |
> |      Clip(zero-shot)      |          85.7\%          |          44.7\%          |          39.9\%          |   |   |          30.9\%          |
> |    Blip2(image caption)   |          61.4\%          |          63.5\%          |          55.5\%          |   |   |          51.8\%          |
> |         LLAVA(VQA)        |          63.8\%          |          71.6\%          |          59.7\%          |   |   |          51.4\%          |
>
>
> The results further demonstrate that our proposed method achieves favorable attack success rates on ViTs. Besides, **our method can well generalized to various models across different vision tasks**, emphasizing the broader applicability and effectiveness of our proposed resolution attack framework. We have incorporated these experiments in the Appendix E.

---

> > ### Comment · Reviewer_XjQe · 2024-11-27
> >
> > Thanks for the response. I think the authors have addressed my concerns, so I raise my score.

---

> > > ### Author Response · Authors · 2024-11-27
> > > **Response to the further feedback of Reviewer XjQe**
> > >
> > > We are glad to hear that you are satisfied with our response. Again, thank you very much for your insightful comments and thoughtful suggestions, especially regarding the inclusion of comparisons with random compression baseline, and discussions on the effectiveness of our method with vision transformers.

---

### Official Review · Reviewer_dpTx · 2024-11-03

**Soundness:** 3
**Presentation:** 3
**Contribution:** 3
**Rating:** 8
**Confidence:** 4

**Summary:**

This paper introduces a novel "resolution attack" that deceives both classifiers and human observers by creating images with distinct semantics at different resolutions. The proposed automated framework generates dual-representation images in a zero-shot manner, using generative priors from large-scale diffusion models and a staged approach to achieve smooth transitions across resolutions.

**Strengths:**

1. The designed "resolution attack" is very innovative and presents interesting effects.
2. It reveals the vulnerability of classifiers under the "resolution attack."
3. The writing is clear, and the ideas are easy to follow.

**Weaknesses:**

1. The classifiers tested in the article are primarily based on CNN methods and do not include evaluations of the latest classifiers. I am curious about how transformer architecture models perform under the "resolution attack," and whether methods using feature pyramids could better overcome the "resolution attack." The authors could provide more analysis in this area.
2. Although the authors validate the effectiveness of their method on existing classifiers, they do not provide a detailed comparison with other attacks (such as adversarial attacks). The authors might consider analyzing how the proposed "resolution attack" compares to other generative attacks that also utilize the diffusion model.
3. The paper touches on applications like face swapping and camouflage without addressing the potential ethical implications.

**Questions:**

I am curious about the performance of transformer architecture classifiers in comparison to CNN architecture classifiers under the "resolution attack," and whether methods using feature pyramids can better overcome the "resolution attack."

---

> ### Author Response · Authors · 2024-11-24
> **Response to Reviewer dpTx (1/2)**
>
> Thank you for taking the time to review our paper, and for your helpful comments. Below we will address your concerns point by point:
>
> > Weakness1 & Question 1: The classifiers tested in the article are primarily based on CNN methods and do not include evaluations of the latest classifiers. I am curious about how transformer architecture models perform under the "resolution attack," and whether methods using feature pyramids could better overcome the "resolution attack." The authors could provide more analysis in this area.
>
> Thanks for the valuable question. We have conducted our proposed Resolution Attacks on Transformer-based models (ViT-B and ViT-L) and Feature Pyramid Network (FPN) methods (Faster RCNN and Mask RCNN). Besides, according to suggestion provided by Reviewer 6DtR, we have evaluated the effectiveness on various vision-language models, including image captioning models (BLIP), Visual Question Answering models (LLAVA) and zero-shot classification models (CLIP). The results are shown in the following tables:
>
>      Additional quantitative results of the Resolution Attack
> |        Classifiers        |                          |      Labeled Attack      |                          |   |   |     Unlabeled Attack     |
> |:-------------------------:|:------------------------:|:------------------------:|:------------------------:|---|---|:------------------------:|
> |                           | $\mathrm{Acc_H}\uparrow$ | $\mathrm{Acc_L}\uparrow$ | $\mathrm{ASR_C}\uparrow$ |   |   | $\mathrm{Acc_L}\uparrow$ |
> |          ViT-b32          |          55.7\%          |          58.1\%          |          41.3\%          |   |   |          83.2\%          |
> |          ViT-l32          |          50.3\%          |          65.3\%          |          44.1\%          |   |   |          87.2\%          |
> | Fasterrcnn\_resnet50\_fpn |            ——            |          47.7\%          |            ——            |   |   |          19.4\%          |
> |  Maskrcnn\_resnet50\_fpn  |            ——            |          57.5\%          |            ——            |   |   |          29.1\%          |
> |      Clip(zero-shot)      |          92.1\%          |          34.9\%          |          33.3\%          |   |   |          62.8\%          |
> |    Blip2(image caption)   |          72.1\%          |          58.7\%          |          53.7\%          |   |   |          91.2\%          |
> |         LLAVA(VQA)        |          71.8\%          |          68.9\%          |          63.6\%          |   |   |          90.0\%          |
>
>      Additional quantitative results of the Resolution Attack with Source image
> |        Classifiers        |                          |      Labeled Attack      |                          |   |   |     Unlabeled Attack     |
> |:-------------------------:|:------------------------:|:------------------------:|:------------------------:|---|---|:------------------------:|
> |                           | $\mathrm{Acc_H}\uparrow$ | $\mathrm{Acc_L}\uparrow$ | $\mathrm{ASR_C}\uparrow$ |   |   | $\mathrm{Acc_L}\uparrow$ |
> |          ViT-b32          |          44.9\%          |          74.5\%          |          56.6\%          |   |   |          58.0\%          |
> |          ViT-l32          |          36.9\%          |          83.9\%          |          60.7\%          |   |   |          62.4\%          |
> | Fasterrcnn\_resnet50\_fpn |            ——            |          53.5\%          |            ——            |   |   |          11.8\%          |
> |  Maskrcnn\_resnet50\_fpn  |            ——            |          67.8\%          |            ——            |   |   |          18.6\%          |
> |      Clip(zero-shot)      |          85.7\%          |          44.7\%          |          39.9\%          |   |   |          30.9\%          |
> |    Blip2(image caption)   |          61.4\%          |          63.5\%          |          55.5\%          |   |   |          51.8\%          |
> |         LLAVA(VQA)        |          63.8\%          |          71.6\%          |          59.7\%          |   |   |          51.4\%          |
>
> The results further demonstrate that our proposed RA can also successfully attack ViTs and FPN methods, exhibiting robustness across various architecture. Besides, our method can well generalized to various models across different vision tasks. This is not surprising as our method is model-agnostic and focus on the semantic of the image. We have incorporated these experiments in the Appendix E.

---

> ### Author Response · Authors · 2024-11-24
> **Response to Reviewer dpTx (2/2)**
>
> > Weakness2: Although the authors validate the effectiveness of their method on existing classifiers, they do not provide a detailed comparison with other attacks (such as adversarial attacks). The authors might consider analyzing how the proposed "resolution attack" compares to other generative attacks that also utilize the diffusion model.
>
> Thanks for your advice. We have no doubt that the reviewer understands the fundamental difference in our proposed Resolution Attack (RA) and other attacks. We provide the detailed comparison here and also supplement these comparison in Appendix D.
>
> We would like to emphasize that the Resolution Attack is fundamentally different from other attacks such as traditional adversarial attacks and other generative attacks utilizing diffusion models:
>
> **First, RA shares very different goals from other attacks.** The Resolution Attack aims to deceive both classifier and human vision while other attacks are mainly focused on machine learning model. Due to this, our method enjoys a natural merit to generalize across different architecture and different vision tasks while other attacks may be easily overfitted to inductive bias of a specific model or vision task.
>
> **Second, RA shares very different technical roadmaps from other attacks.** RA aims to synthesize dual-semantic images and focus more on the content of the images. Thus, RA requires to search a broader landscape of the image manifold and heavily relied on the semantic-level understanding of the images. (We achieve this by utilizing the strong generative priors of the large-scaled Stable Diffusion.) However, other attacks only involve pixel-level perturbations and search a p-ball around the target image.
>
> Furthermore, the primary objective of RA is to generate images with distinct semantics across different resolutions. This differs fundamentally from traditional adversarial attacks and generative attacks based on diffusion models. As the first work to propose the concept of resolution attacks, we establish a baseline for this novel direction.
>
> > Weakness3: The paper touches on applications like face swapping and camouflage without addressing the potential ethical implications.
>
> Thank you for highlighting the need to address ethical considerations. **We have now explicitly discussed these aspects in of the revised paper (Appendix G). Specifically, we emphasize that our goal is to explore vulnerabilities in current vision models.** Though our approach is under the risk of malicious purpose, our generated image can be employed to fine-tune existing classifiers, further improving their robustness. Besides, the generated images can also be detected by deepfake detectors. In summary, our study seeks to advance understanding in model robustness while advocating for responsible and ethical use of such methodologies in the broader research community.

---

> ### Comment · Reviewer_dpTx · 2024-11-27
>
> Thanks for the author's response. Most of my concerns have been addressed, so I will increase my rating to 8.

---

> ### Author Response · Authors · 2024-11-27
> **Thank you for your recognition of our work**
>
> We are delighted to have addressed your concerns and appreciate your constructive insights, which have significantly enhanced the quality of our paper. Specifically, your recommendation to analyze the performance of Transformer-based models under the resolution attack and to explore methods utilizing feature pyramids has helped us better demonstrate the broader applicability and robustness of our approach. Additionally, your other suggestions greatly contributed to enriching the depth of our work.
>
> We deeply appreciate your feedback and the time you took to review our paper. Thank you once again for your support and for raising the rating of our work.

---

### Official Review · Reviewer_2GCH · 2024-11-04

**Soundness:** 2
**Presentation:** 2
**Contribution:** 2
**Rating:** 6
**Confidence:** 4

**Summary:**

Ensuring model robustness is critical for the stability and reliability of machine learning systems. Although substantial research has addressed areas such as adversarial and label noise robustness, robustness against variations in image resolution—especially high-resolution images—remains underexplored. To bridge this gap, this work introduces a novel form of attack: the resolution attack. This attack generates images that display different semantics at varying resolutions, deceiving both classifiers and human observers. The resolution attack is implemented through an automated framework designed to create dual-semantic images in a zero-shot manner. Specifically, it leverages large-scale diffusion models for their comprehensive image construction capabilities and employs a staged denoising strategy to achieve smooth transitions across resolutions. Using this framework, resolution attacks were conducted on several pre-trained classifiers, with experimental results demonstrating a high success rate. These findings validate the effectiveness of the proposed framework and highlight the vulnerability of current classifiers to resolution variations. Additionally, this framework, integrating features from two distinct objects, presents a powerful tool for applications like face swapping and facial camouflage.

**Strengths:**

1. This paper introduces a new test-time attack, the resolution attack, which demonstrates effective performance across several classifiers.

**Weaknesses:**

1. The attack scenario presented is quite limited. The paper uses a high resolution of $512 \times 512$, but the size of low-resolution images is unclear, though figures suggest it might be $32 \times 32$. In practice, this resolution is only common in classifiers trained on small-scale datasets like MNIST or CIFAR-10/100, whereas real-world classifiers, such as those trained on ImageNet, typically use a $224 \times 224$ input resolution, making this type of attack less applicable.

2. It is expected that this attack could succeed, as adversarial images here are generated from scratch. However, in practical scenarios, adversarial images are typically crafted by modifying an uploaded clean image to influence predictions while maintaining visual similarity to the original. Generating adversarial images from scratch, therefore, has limited practical application.

3. The module proposed lacks novelty, as the separation of control between modules for detail and shape generation is quite standard.

4. Key implementation details are missing, such as the resolutions at which images successfully achieve adversarial effects.

**Questions:**

See weakness above.

---

> ### Author Response · Authors · 2024-11-24
> **Response to Reviewer 2GCH (1/3)**
>
> Thank you for your valuable feedback. Below, we will respond to each of your comments in detail, addressing specific issues while ensuring our reasoning is clear and fully responsive to your concerns.
>
> > Weakness 1: The attack scenario presented is quite limited. The paper uses a high resolution of 512×512, but the size of low-resolution images is unclear. Figures suggest it might be 32×32. In practice, this resolution is only common in classifiers trained on small-scale datasets like MNIST or CIFAR-10/100, whereas real-world classifiers, such as those trained on ImageNet, typically use higher input resolutions, making this type of attack less applicable.
>
> Thank you for this comment. We respectfully disagree with the reviewer on these comments that the practicity of this research is limited. We first want to clarify that we leverage **low-resolution images of 64x64** (not 32x32) by downscaling the high resolution images with a factor of 3, as we have stated in the original submission (Appendix A). Besides, we further want to clarify that **the classifiers** we utilize is for **224x224** images, not for 32x32 images. We do this to simulate the real-world scenarios in which classifiers encounter reduced-resolution inputs.
>
> We further note that our proposed resolution attack is highly applicable and relevant in broader and practical settings due to the prevalence of low resolution images in various domains. Specifically, we identify several application scenarios where low-resolution images are commonly used:
>
> + **Compressed Internet Images**: Images uploaded to social media or online platforms often undergo heavy compression, reducing their resolution significantly. This creates opportunities for resolution attacks to exploit classifiers processing such images.
>
> + **Autonomous Driving Systems**: In autonomous vehicles, small and distant objects (e.g., traffic signs or pedestrians) often appear as low-resolution entities. Resolution attacks could raise fresh safety concerns in these systems.
>
> + **Surveillance Systems**: Surveillance cameras often prioritize storage efficiency by capturing video footage at lower resolutions. Resolution attacks could compromise object detection systems designed for such footage.
>
> + **IoT Devices**: Many IoT devices, such as smart home cameras or wearable devices, operate with constrained hardware and capture images at reduced resolutions. These systems are particularly susceptible to resolution-based vulnerabilities.
>
> Given this, we believe our proposed resolution attack is highly applicable and will influence a wide range of field.

---

> > ### Comment · Reviewer_2GCH · 2024-11-24
> > **Comments to the Authors**
> >
> > Thank you for the feedback!
> >
> > Regarding this point, the primary concern is that the image is downsampled from $512 \times 512$ to $64 \times 64$, while the classifier expects an input of $224 \times 224$. It’s unclear whether the downsampled images are directly fed into the classifier or upsampled back to $224 \times 224$. This aspect needs clarification and justification. In addition, it is possible to conduct some experiments on classifiers with default $64 \times 64$ inputs.
> >
> > As for the application scenario you mentioned, it is more practical for the generated adversarial images to maintain clear visual similarity to the original, clean images. For this reason, I would prefer an approach based on additive noise. Does your method support this additive approach?

---

> ### Author Response · Authors · 2024-11-24
> **Response to Reviewer 2GCH (2/3)**
>
> > Weakness 2: Generating adversarial images from scratch has limited practical application. In practical scenarios, adversarial images are typically crafted by modifying uploaded clean images to influence predictions while maintaining visual similarity to the original.
>
> Maybe there is a misunderstanding. **First, our method also support crafting uploaded lean images (RAS, Section 3.2&4.3). Second, our goal is completely different from the adversarial examples**: we aim to generate images with dual labels to influence predictions in both high and low resolution, while adversarial examples only aim to influence the prediction. Additionally, adversarial attacks often overfit specific model architectures or inductive biases of particular tasks, limiting their generalization. In contrast, resolution attacks naturally generalize across diverse architectures and visual tasks. Thus, the resolution attacks is more challenging than adversarial attacks. Below, we clarify the difference between resolution attacks and adversarial attacks in detail.
>
> **Comparison with Traditional Adversarial Attacks:**
> Traditional adversarial attacks focus on modifying clean images minimally, aiming to deceive classifiers without altering the image's apparent semantics. In contrast, our resolution attacks generates images with dual semantics. This innovative semantic duality extends beyond merely flipping labels, as it introduces a new dimension of misclassification by targeting classifiers across resolution scales. Furthermore, our method inherently avoids overfitting to specific architectures or tasks. While adversarial examples are prone to exploiting inductive biases of particular models or visual tasks, resolution attacks leverage generative capabilities to **generalize naturally across various architectures and tasks**. We conducted experiments on various CNN architectures, including different frameworks (ViTs, FPN), and across a range of tasks (VQA, image captioning, zero-shot classification). We have incorporated these experiments in the Appendix E. The experimental results underscore the broad applicability of our attack.
>
> **Usefulness and Applicability**:
> The dual-semantic capability and source-guided design of RA/RAS enhance its practical relevance in auditing generative models, evaluating robustness, and detecting vulnerabilities in classifiers. Structural consistency with the source image is quantified using the SSIM (Structural Similarity Index) metric, demonstrating that the generated images remain contextually plausible.
>
> By addressing these differences and demonstrating the unique value of our approach, we emphasize that RA and RAS offer significant advancements in adversarial attack methodologies, paving the way for novel research and applications.

---

> > ### Comment · Reviewer_2GCH · 2024-11-24
> > **Comments to the Authors**
> >
> > The primary concern here is that the generated images, even those in Figure 2(b), lack visual similarity to the source images. Additionally, unlike adversarial attacks, there is no clearly defined bound for the perturbations.

---

> ### Author Response · Authors · 2024-11-24
> **Response to Reviewer 2GCH (3/3)**
>
> > Weakness 3: The module proposed lacks novelty, as the separation of control between modules for detail and shape generation is quite standard.
>
> Thanks for this comment. We would like to elaborate on the novelty of our framework:
>
> **The core innovation of our work lies in proposing a completely new type of attack—Resolution Attacks (RA)**—which systematically exploits the resolution discrepancy between high-resolution and low-resolution image classifiers. This attack concept is novel and unprecedented in the literature.
>
> To operationalize this, **we designed a dual-stream denoising network, which ensures the precise alignment of dual-semantic representations at high resolution (PH) and low resolution (PL).** This network is tailored to the specific requirements of RA, offering fine-grained semantic control to simultaneously generate distinct, yet semantically aligned outputs for PH and PL.
>
> Additionally, when using the controlnet module in the RAS, **our framework further extends flexibility by using different control conditions** to adapt to the semantic gap between PH and PL. This ensures more targeted generation of dual-representation images, significantly improving the practicality.
>
> > Weakness 4: Key implementation details are missing, such as the resolutions at which images successfully achieve adversarial effects.
>
> We understand your concern about implementation details. About "the resolutions at which images successfully achieve adversarial effects", we have mentioned in the paper. This information is provided in the original submission (Appendix A), where we specify a downsampling factor of 3, resulting in low-resolution images of size 64×64.
>
> We hope the above response can fully address your queries and demonstrate the scientific rigor and research potential of our work.

---

> ### Author Response · Authors · 2024-11-25
> **Response to the further feedback of Reviewer 2GCH (1/2)**
>
> > Feedback1.1: Regarding this point, the primary concern is that the image is downsampled from 512 * 512 to 64 * 64, while the classifier expects an input of 224 * 224. It’s unclear whether the downsampled images are directly fed into the classifier or upsampled back to 224 * 224. This aspect needs clarification and justification. In addition, it is possible to conduct some experiments on classifiers with default 64 * 64 inputs.
>
> Thank you for the feedback. Regarding the concern about the downsampling process, we confirm that the downsampled images (64×64 resolution) are directly fed into the classifier. The classifier preprocesses these images using PyTorch's default pipeline, as described in its documentation:
>
> "Accepts PIL Image, batched (B, C, H, W) and single (C, H, W) image torch.Tensor objects. The images are resized to resize_size=[232] using interpolation=InterpolationMode.BILINEAR, followed by a central crop of crop_size=[224]. Finally, the values are first rescaled to [0.0, 1.0] and then normalized using mean=[0.485, 0.456, 0.406] and std=[0.229, 0.224, 0.225]."
>
> Furthermore, in response to your inquiry regarding the "classifiers with default 64 * 64 inputs," as there is currently no classifier specifically tailored for 64×64 resolution inputs, we conducted supplementary experiments using a model trained on the CIFAR-10 dataset (32 * 32) [1]. The experimental results are presented in the table below.
>
> |      | RA-labeled | RA-unlabeled | RAS-labeled | RAS-unlabeled |
> |:----:|:----------:|:------------:|:-----------:|:-------------:|
> | $\mathrm{Acc_L}\uparrow$ |    84.5%   |     82.9%    |    93.8%    |     74.7%     |
>
> The results demonstrate the effectiveness of our approach even for low-resolution classifiers.
>
> [1] https://github.com/chenyaofo/pytorch-cifar-models/tree/master/pytorch_cifar_models

---

> ### Author Response · Authors · 2024-11-25
> **Response to the further feedback of Reviewer 2GCH (2/2)**
>
> > Feedback1.2: As for the application scenario you mentioned, it is more practical for the generated adversarial images to maintain clear visual similarity to the original, clean images. For this reason, I would prefer an approach based on additive noise. Does your method support this additive approach?
>
> > Feedback2: The primary concern here is that the generated images, even those in Figure 2(b), lack visual similarity to the source images. Additionally, unlike adversarial attacks, there is no clearly defined bound for the perturbations.
>
> Thank you for your comments. Your main focus is on the issues of the "visual similarity to the source images" and "bound for the perturbations". Since both Q1.2 and Q2 share a common focus, we address them together below.
>
> Our approach aims to assess the semantic robustness of classifiers at different resolutions by generating images using the capability of generative models to create images with dual semantics. Maintaining strict visual similarity to the source images is not the primary goal of our method and does not serve as an appropriate criterion for assessing its effectiveness. Instead, the focus of our approach is to explore a broader search space beyond pixel-level perturbations, enabling more comprehensive evaluations of classifier vulnerabilities across diverse models and tasks.
>
> **Methodological Perspective: Semantic-Level Generation vs. Pixel-Level Perturbation**
>
> We emphasize that our method differs fundamentally from traditional adversarial attacks. Conventional adversarial attacks typically introduce imperceptible pixel-level perturbations to the original image, carefully computed to exploit the classifier's decision boundaries. These perturbations often have a well-defined bound (perturbation bound), which plays a critical role in assessing the classifier's robustness to adversarial noise.
>
> However, modern classifiers are increasingly exposed to a variety of generative model outputs, including high-resolution synthesized images. Perturbation bounds, while relevant for pixel-level perturbations, constrain the search space of adversarial samples and may not provide a comprehensive evaluation of classifier robustness.
>
> Our resolution attack operates at the semantic level, focusing on generating images that leverage the generative model's priors and semantic representations. This enables the method to explore a broader search space beyond the constraints of pixel-level perturbation bounds. Moreover, our method is not tailored to any specific classifier, making it more versatile for evaluating robustness across different models and tasks.
>
> **Visual Similarity and Semantic Gap**
>
> Regarding your concerns about the visual similarity of the generated images to the source images, it is important to note that the visual similarity depends on the semantic gap between the high-resolution prompt and the low-resolution prompt. When the semantic gap is small, as demonstrated in Figures 17 and 18 of the paper, the generated images exhibit a much closer resemblance to the source images.
>
> While additive noise-based approaches are effective for preserving high visual fidelity, they inherently remain limited to the pixel level. In contrast, our approach leverages the power of generative models to synthesize images that align with the semantic attributes of the prompts, enabling a novel perspective for evaluating classifier robustness.
>
> We hope the above explanation addresses your concerns and we are open to answering any further questions you may have.

---

> > ### Comment · Reviewer_2GCH · 2024-11-27
> > **Comments to the Authors**
> >
> > I don’t have significant issues at this point. However, I recommend that the authors revise the draft to provide more detailed illustrations of the practical scenario and experimental setup, including specifications such as sizes. Additionally, it would be better to use proper citations for published papers instead of referencing arXiv numbers.
> >
> > I may be willing to raise my score from 5 to 6 if all the concerns mentioned above are adequately addressed.

---

> > > ### Author Response · Authors · 2024-11-27
> > > **Response to the further feedback of Reviewer 2GCH**
> > >
> > > We are very pleased to hear that you have no significant issues at this point and appreciate your willingness to reconsider your score. Thank you for your thoughtful and constructive suggestions. We will carefully address the revisions you suggested, updating the draft to include the necessary details and proper citations. The revised version will be uploaded as soon as possible. Once again, thank you for your detailed feedback and consideration.

---

> > > ### Author Response · Authors · 2024-11-27
> > > **The revised manuscript has been uploaded!**
> > >
> > > Thank you once again for your valuable feedback, which has greatly enhanced the clarity and quality of our manuscript. We have taken your advice into consideration and have revised the manuscript accordingly. The revised sections are highlighted in orange for your convenience.
> > >
> > > + We have provided a more detailed description of the practical scenario of our attack. Specifically, we have emphasized the significance of robustness towards low-resolution images and provided corresponding practical examples (Lines 49-60).
> > >
> > > + We have expanded our explanation of the experimental setups. Specifically, we have clarified the resolution (64x64) and preprocessing pipeline (first rescale, then central crop) for the low-resolution images (Lines 400-413). Additionally, we have specified the resolutions of the target classifiers (Lines 316-318). We believe that incorporating these details will improve the overall clarity of our draft.
> > >
> > > + We have thoroughly reviewed the Reference and updated the citations of the published papers as per your suggestions.
> > >
> > > We are deeply grateful for your insightful comments and the time you have dedicated to reviewing our manuscript!

---

> > > ### Author Response · Authors · 2024-11-30
> > > **Thank you for your recognition of our work**
> > >
> > > We sincerely thank you for providing a higher score for our work. Your meticulous and professional feedback has been invaluable in improving our manuscript. In particular, your recommendation to provide more detailed descriptions of the practical scenarios and experimental setups, along with the suggestion to replace arXiv numbers with proper citations, has greatly enhanced the clarity and thoroughness of our work.
> > >
> > > Once again, we deeply appreciate your patience and thoughtful review. Your constructive comments have played a key role in perfecting our manuscript, and we are sincerely grateful for your support and guidance.

---

### Author Response · Authors · 2024-11-24
**General Response**

We are sorry for this late response. We express our gratitude to all the reviewers for their insightful and constructive feedback. Each reviewer's comments have been individually addressed and responded to.

Additionally, we have uploaded the revised manuscript, incorporating the following key revisions:

+ We evaluate our proposed Resolution Attack on various architecture (ViT, FPN) and across various vision tasks (VQA, Caption), which further demonstrate the effectiveness of RA and reflect the model-agnostic and task-agnostic merit of RA (Appendix E).

+ We provide detailed comparisons to current attack methods, particularly adversarial attacks, to clarify the position of our work within the current arts (Appendix D).

+ We supplement an ethical statement to address potential concerns (Appendix G).

Lastly, we would appreciate all reviewers' time again. We sincerely hope that our response would address the reviewers' concerns. Further feedback and discussions are appreciated.

---

### Meta-Review · Area_Chair_TLLP · 2024-12-20

**Metareview:**

This paper presents a novel attack method called resolution attack, which creates images that display different semantic content when viewed at varying resolutions. These dual-representation images can successfully mislead both machine-learning classifiers and human observers. The automated framework leverages generative priors from large-scale diffusion models to produce these images in a zero-shot fashion, employing a staged optimization approach to ensure smooth transitions across resolution levels. The reviewers agree the work has the following advantages: (1) The attack is novel and interesting; (2) It reveals the vulnerability of classifiers under the attack.  (3) The writing is clear, and the ideas are easy to follow. They also raise some questions: (1) The attack scenario presented is quite limited with limited practical application for generating images from scratch. (2) Unclear introductions for the implementations. (3) Lack the evaluation on the latest classifier. (4) Lack of some baseline methods. Most of the concerns are addressed. However, the final version is suggested to revise as follows: (1) Providing more detailed illustrations of the practical scenario and experimental setup. (2) Adding new results in the rebuttal. (3) Revising the paper according to the suggestions of reviewers, e.g., the publication formats.

**Additional Comments On Reviewer Discussion:**

All reviewers provide solid comments and feedback on the authors' responses. Most of the concerns are addressed after rebuttal. All reviewers rate positive scores.

---

### Decision · Program_Chairs · 2025-01-22

Accept (Poster)